# Bacteria break through one-micrometer-square passages by flagellar wrapping

Aoba Yoshioka[1], Yoshiki Y. Shimada[2], Toshihiro Omori[3], Naoki A. Uemura [1], Kazutaka Takeshita [4], Kota Ishigami[5], Hiroyuki Morimura [5], Maiko Furubayashi[5], Tetsuo Kan [2] ✉, Hirofumi Wada [6] ✉, Yoshitomo Kikuchi [5] ✉ & Daisuke Nakane [1] ✉

Confined spaces are omnipresent in the micro-environments, including soil aggregates and intestinal crypts, yet little is known about how bacteria behave under such conditions where movement is challenging due to spatial confinement that limited effective diffusion. Stinkbug symbiont *Caballeronia insecticola* navigates a narrow gut passage about one micrometer in diameter to reach the stinkbug's symbiotic organ. Here, we developed a microfluidic device mimicking the host's sorting organ, wherein bacterial cells are confined in a quasi-one-dimensional fashion, and revealed that this bacterium wraps flagellar filaments around its cell body like a screw thread to control fluid flow and generate propulsion for smooth and directional movement in narrow passages. Physical simulations and genetic experiments revealed that hook flexibility is essential for this wrapping; increasing hook rigidity impaired both wrapping motility and infectivity. Thus, flagellar wrapping likely represents an evolutionary innovation, enabling bacteria to break through confined environments using their motility machinery.

Bacteria live everywhere on earth, from deep-ocean trenches to the tops of mountains, and even inside animals and plants. The microhabitats that bacteria inhabit are replete with confined environments such as small pores, cavities, and narrow channels[1]. Soil, for example, is a mixture of particles of various sizes at the micron scale, which creates complex confined spaces[2]. Interspaces between cells and microvilli are also confined[3], where bacteria inhabit or invade. In recent years, bacterial behavior in micro-environments has been studied in some model species by using microdevices that mimic their natural habitats[4]. However, it remains largely unknown how environmental bacteria with diverse styles of motility behave in confined environments.

Bacterial motility adapts to fluid dynamics at low Reynolds number, where inertia is negligible and viscous forces are dominant[5]. Moreover, in confined spaces, the apparent viscosity increases dramatically at the solid-liquid interface due to interactions between materials at a boundary. From this physical viewpoint, moving in micrometer confined spaces, which bacteria often encounter in their natural habitats, is thought to be challenging for bacteria, like rock excavation.

A few notable examples of confined-space niches encountered by bacteria are found in animals, such as squids and insects, which sort specific bacterial symbionts from environmental microbiota by using a constricted region (CR) in front of their symbiotic organs. Such a sorting gate has been found in the squid-*Vibrio* luminescent symbiosis[6], and recently well-described in the bean bug *Riptortus pedestris* which harbors with *Caballeronia insecticola* in its gut[7,8]. The

[1]Department of Engineering Science, Graduate School of Informatics and Engineering, The University of Electro-Communications, Tokyo, Japan. [2]Department of Mechanical Engineering and Intelligent Systems, Graduate School of Informatics and Engineering, The University of Electro-Communications, Tokyo, Japan. [3]Department of Finemechanics, Tohoku University, Sendai, Japan. [4]Department of Biotechnology, Faculty of Bioresource Sciences, Akita Prefectural University, Akita, Japan. [5]Biomanufacturing Process Research Center, National Institute of Advanced Industrial Science and Technology (AIST), Sapporo, Japan. [6]Department of Physical Sciences, Ritsumeikan University, Shiga, Japan. ✉e-mail: tetsuokan@uec.ac.jp; hwada@fc.ritsumei.ac.jp; y-kikuchi@aist.go.jp; dice-k@uec.ac.jp

structure of these sorting organs is very simple, a tubular structure with a lumen of one micrometer in diameter and a few hundred micrometers in length[9,10], but is remarkably effective at sorting out the hosts' specific symbionts only. In both cases, the symbionts described above possess polar flagella, which are crucial for crossing their host's CR, a narrow passage filled with a mucus-like matrix[10,11]. While bacteria commonly swim by rotating helical flagellar filaments, which trail the cell body[12], recent studies have demonstrated a novel mode of swimming in these symbiotic bacteria wherein symbionts wrap flagellar filaments around the cell body and propel themselves with the rotary motor positioned at the front of the cell[13,14]. Since flagellar wrapping is known to be effective in moving in viscous conditions and escaping substrate surfaces[15], this unique swimming mode is thought to play a pivotal role in adapting to the internal environments of the hosts[3]. Little is known, however, about the ecological significance of flagellar wrapping in spatially-confined narrow passages.

Here, we show that flagellar wrapping enables bacteria to traverse micrometer-scale confined passages. Using a quasi-one-dimensional microfluidic device that mimics the host's CR, we combine in vivo imaging, high-speed fluorescence, and hydrodynamic modeling to reveal that flagellar wrapping is pivotal for passing through the CR, and that confinement promotes wrapped configurations that enhance propulsion. Genetic manipulation of hook flexibility demonstrates its essential role in wrapping, and

bacterial species-level performance in confinement correlates with host infection efficiency.

## Results

### Flagellar wrapping in the host's sorting organ

We first investigated how the insect symbiont, *C. insecticola*, passes through the sorting organ of its host, *R. pedestris* (Fig. 1a,b). A second instar nymph was dissected after being fed *C. insecticola* expressing a green fluorescent protein (GFP), and the digestive tract was observed under optical microscopy around the entry gate of the symbiotic organ, CR (Fig. 1c,d). *C. insecticola* cells located by GFP signal had crossed the CR and were observed arranged in a row at the subsequent narrow passage connecting the M4B and M4 midgut regions 2-4 hours after feeding. Time-lapse imaging revealed that the cells showed repeated back-and-forth movements, but moved directionally from the M3 to M4 regions through the CR with a net displacement of 50 μm min$^{-1}$ (Fig. 1e and Movie S1), indicating that *C. insecticola* cells can break through the host's narrow passages, a 200 μm distance, in a few minutes. The bean bug *R. pedestris* ingests liquid food that passes through M1, M2, and M3, where nutrients and water are fully absorbed, transported throughout the body via hemolymph, and metabolic wastes are excreted via the Malpighian tubules[10]. The posterior regions, including the CR, are therefore functionally isolated from the digestive flow. Since the CR is an extremely narrow passage through

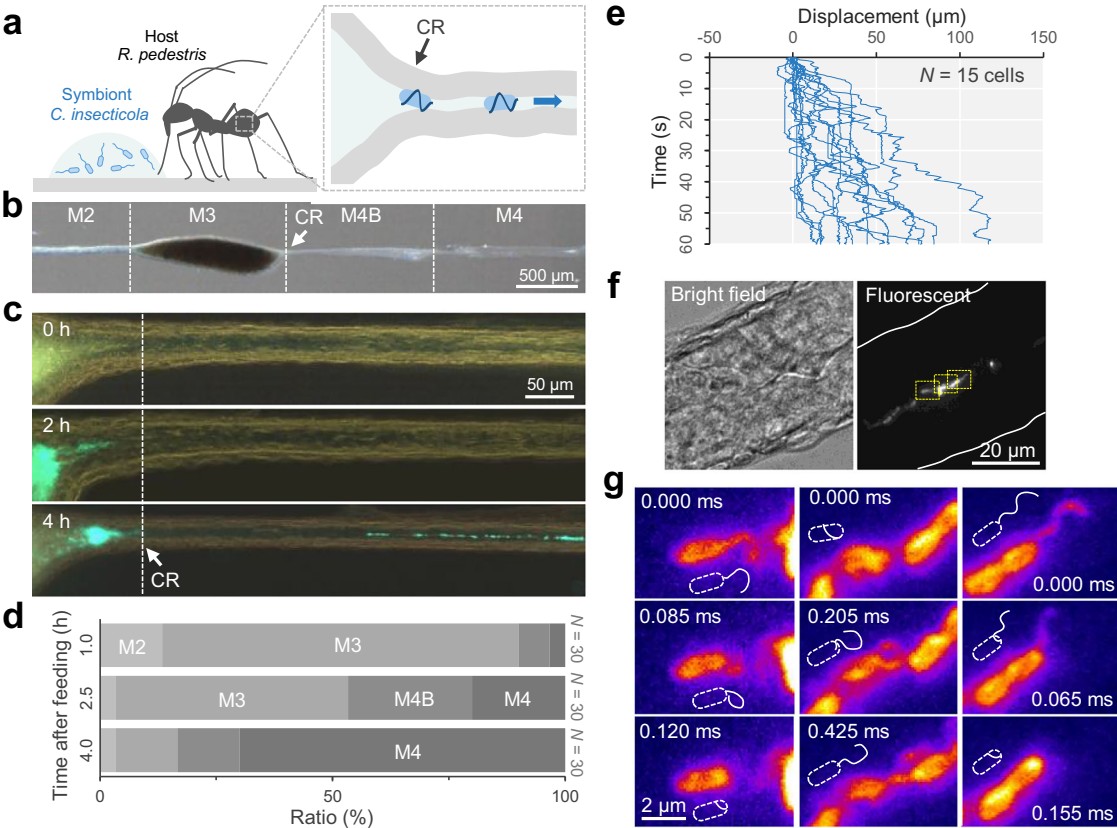

**Fig. 1 | Single cell imaging of flagellar wrapping motility in the gut sorting organ. a** Schematic illustration of a second instar nymph of *R. pedestris* fed with *C. insecticola*. **b** Midgut organization and the bacterial sorting organ (CR). Representative image from three independent experiments showing similar results. **c** Symbiont localization at the CR-M4B region. The images show the gut region 0, 2, and 4 h after feeding of GFP-labeled *C. insecticola*. Note the symbionts start entering the CR 2 h after feeding and are passing through the narrow duct 4 h after feeding. See also Movie S1. **d** Time required for symbiont sorting. Ratio of the locations where the GFP signal from *C. insecticola* was detected at the most distal part is shown (*N* = 30 nymphs). **e** Time course of symbiont displacement in the CR-

M4B region 2–4 h after feeding of GFP-labeled *C. insecticola*. Single-cell movement colored by a blue line, and 15-cell displacements are overlaid. **f** Direct visualization of the flagellar filament of the symbiont in the M4B region. *C. insecticola* cells fluorescently labeled by amine-reactive dye were fed to a second instar nymph, and the gastrointestinal tract was dissected 2–4 h after infection. Left: Bright-field. Right: Fluorescent. Representative result from at least three independent experiments that yielded similar observations. See Movie S2. **g** Flagellar wrapping in the CR-M4B region. Yellow-boxed regions in (**f**) were magnified, and time-lapsed images are presented. Schematic are overlaid to indicate the cell body and flagellar filaments. Source data are provided with this paper.

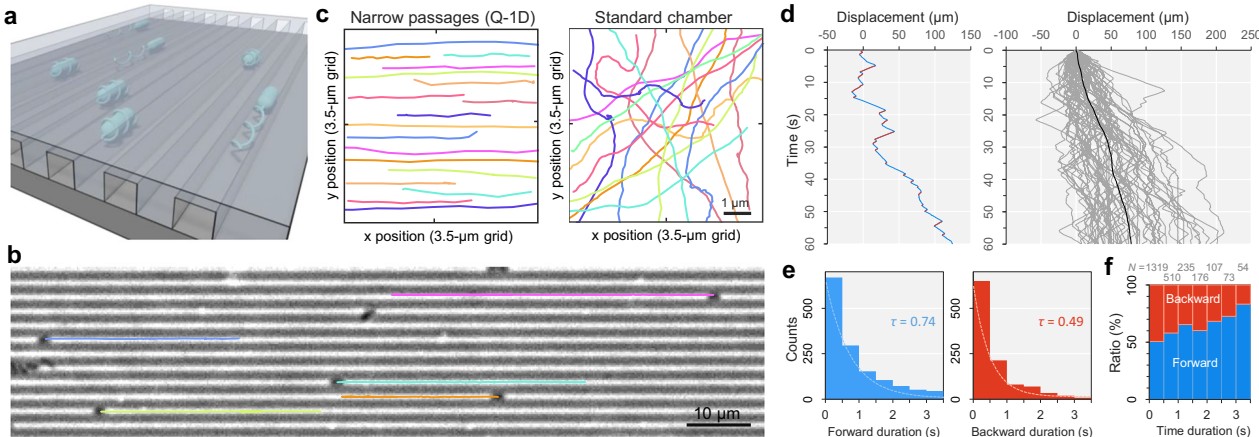

**Fig. 2 | Smooth and directional movement of symbiotic bacteria in microfluidic narrow passages. a** Schematic of confined bacteria in the Q-1D microfluidic device. **b** Smooth and directional movement of *C. insecticola* cells in Q-1D. The cell trajectories over 20 s were overlaid on the phase-contrast image (see Movie S3). **c** Single-cell trajectories for 1 s. Left: Q-1D. Right: Standard chamber. **d** Time course of cell displacement in Q-1D. Left: Single cell. Forward and backward movements were detected as positive and negative displacements along the *x* axis, and presented by the blue and red colored lines, respectively. Right: Overlays of 61 cells and the average. **e** Distribution of the forward and backward duration of swimming time in Q-1D. Duration from one directional change to the next were measured and color-coded as in (**d**). Dashed lines show the fit of single exponential decay, where time constant τ is presented. **f** Duration ration of forward and backward direction. The duration of each run was measured, and the number of forward versus backward runs was counted. The ratio of forward runs was then calculated for runs with durations falling into 0.5 s intervals. The sample number of each bin are presented at the top. Source data are provided with this paper.

which even colored liquid cannot pass, as reported previously[10], we consider that fluid flow is unlikely to occur in this region after dissection. Indeed, no visible flow was observed during our microscopic observations in the narrow passage. Flagellar filaments of *C. insecticola* can be fluorescently labeled with amine-reactive dyes[13]. High-speed imaging of *C. insecticola* with fluorescently-labelled flagellar filaments revealed that *C. insecticola* adopts a wrapped mode of flagellar motility while in the CR-M4B narrow passage (Fig. 1f,g and Movie S2). Notably, fluorescently labeled flagellar filaments were visualized at the front of the cell body during entry into the CR, indicating that cells orient themselves with their flagellar motors facing forward. Since U-turns are physically impossible once inside the narrow passage, this orientation plausibly leads to a directionally biased movement toward the symbiotic organ. This observation supports the idea that flagellar wrapping is important for breaking through the host's sorting organ.

## Our device: quasi-one-dimensional device (Q-1D) mimicking the sorting organ

To observe the symbiont's behavior in micrometer-level narrow passages in more detail, we created a dimethylpolysiloxane (PDMS) microfluidic device with linear, open channels whose width and depth were limited to 1 μm, and filled with a viscous solution of 0.4% methylcellulose (MC) (Fig. S1a,b). In this device, the diffusion of microbeads was almost entirely restricted, with their apparent diffusion coefficient being reduced to 1/60 of that in a standard sample chamber (Fig. S1c–e). Although the exact viscoelastic properties of fluid in the CR are currently unknown, we used MC solution to approximate the physical parameters of the insect's sorting organ. In the bulk liquid, *C. insecticola* cells exhibited repeated directional reversals confirm that the shares the general reversal swimming pattern characteristic of polarly flagellated species (Fig. S2). When *C. insecticola* cells were confined in the device with MC (Fig. 2a), the cells oriented themselves parallelly along the narrow passages and exhibited directional movement (Fig. 2b,c and Movie S3). Since the Q-1D device is structurally symmetric and cells were randomly distributed throughout the channels, we measured the net displacement of individual cells, regardless of whether they moved to the left or right along the channel. In this analysis, the net displacement for 1 min in both directions were treated equally as positive displacements. In

device, the symbiont cells showed switchbacks within a short time period of a few seconds but were biased to one side for a longer period of 1 minute at a net displacement of 80 μm min⁻¹ (Fig. 2d). The time from one directional change to the next showed a distribution of single exponential decay in both forward and backward directions (Fig. 2e), whereas the bias in the forward direction gradually increased with time (Fig. 2f). Considering that the deletion mutant of *cheA*, which continuously rotates their flagellar motor in the counterclockwise (CCW), stopped movement in the device for a long time (Figure S3 and Movie S4), clockwise (CW) rotation of the flagellar motor is involved in directional movement. We used this quasi-one-dimensional device (Q-1D) for the remainder of this study.

## Flagellar wrapping gives an advantage in narrow passages

The Q-1D also allowed us to observe the motility of *Salmonella enterica*, a model species for the study of bacterial flagella (Fig. S4a, Fig. S5a and Movie S5). However, the directionality of their movement in the narrow passage was less clear (Fig. S4b). The time intervals between directional changes in *S. enterica* followed a single exponential decay, with a time constant comparable to that observed in *C. insecticola* (Fig. S4c), whereas the forward directional bias remained nearly constant over time (Fig. S4d). In contrast, *Vibrio fischeri* (*Aliivibrio fischeri*), the luminescent symbiont of the Hawaiian bobtail squid *Euprymna scolopes*[6], moved smoothly in Q-1D (Fig. S5a and Movie S6). Remarkably, the average of mean square displacement (MSD) of *V. fischeri* and *C. insecticola* in Q-1D showed parabolic curve, as well as that of *C. insecticola* in CR-M4B narrow passage in host, indicating directional movement (Fig. S5b). This difference may be caused by the arrangement of the motility machinery, i.e., multiple flagella randomly distributed over the entire cells in *Salmonella*, while a single or few flagella extend from one end of the cell in *C. insecticola*, and *V. fischeri*, respectively[15], or due to the lack of flagellar wrapping in *Salmonella*. In the Q-1D device, the signed mean displacement of *C. insecticola*, *Salmonella*, and *V. fischeri* moved symmetrically around zero, confirming that their motion was isotropic and not biased by chemotaxis or any other gradient (Fig. S5c). To further investigate the importance of polar flagella in navigating confined spaces, we observed motility in bacterial species closely related to *C. insecticola* and possessing polar flagella using the Q-1D device.

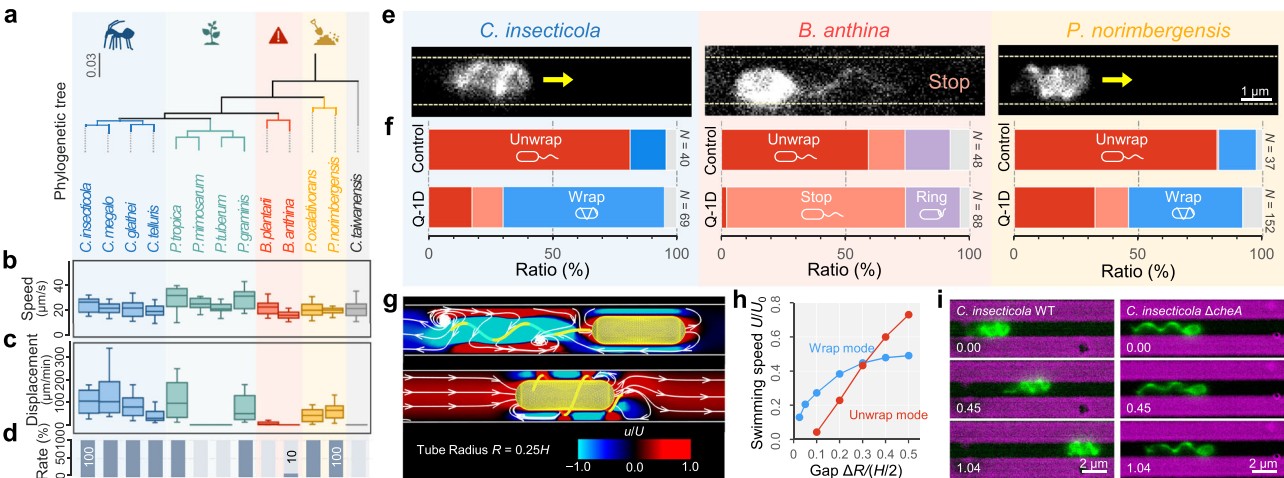

**Fig. 3 | Interspecies variation in flagellar wrapping and its advantages in confined spaces. a** Phylogenetic tree of *Burkholderia* sensu lato group based on 16S ribosomal RNA sequences. Stinkbug-associated *Caballeronia*, plant-associated *Paraburkholderia*, pathogenic *Burkholderia* sensu stricto group, and *Pandoraea* species are highlighted with blue, green, red, and yellow, respectively[7]. **b** Swimming speed for 1.0 s in growth medium for each species. **c** Cell displacement for 1 min in Q-1D. Box plots in (**b**–**c**) represent the minimum, maximum, sample median, and the first and third quartiles (*N* = 20 cells). **d** Infection ratio of the bean bug *R. pedestris* (data derived from our previous paper[7]). **e** Fluorescently labeled flagellar filament of *C. insecticola*, *B. anthina*, and *P. norimbergensis* in Q-1D. The edge of the narrow passage is presented by dashed yellow lines. The arrows indicate the moving direction of each cell. See Movie S10. **f** The fraction of flagellar filament morphology in growth medium and Q-1D. The wrapped, unwrapped, incompletely wrapped, and uncategorized flagella are colored by red, purple, and, blue, respectively. Gray indicates uncategorized flagella that could not be clearly assigned to any of the defined categories. **g** Numerical calculations of time-averaged flow field around the bacterial model in a narrow tube. Left: Unwrapped mode. Right: Wrapped mode. White arrows are the streamline and contour color indicate the velocity component in the swimming direction normalized by the swimming speed *U*. **h** Swimming speed *U* as a function of the gap in the cylinder tube Δ*R*. The values are normalized by the swimming velocity of the unwrapped mode in the free space $U_0$ and a half body length *H*/2, respectively. The gap width between was defined geometrically and systematically varied in the numerical simulations to assess its effect on propulsion. **i** Snapshot of cell behavior in Q-1D. Fluorescently labeled flagellar filament of *C. insecticola* WT and Δ*cheA*. See Movie S13. Source data are provided with this paper.

In total, 13 species of *Burkholderia* sensu lato and allied groups, including members of the genera *Caballeronia*, *Paraburkholderia*, *Burkholderia*, *Pandoraea*, and *Cupriavidus*, were compared (Fig. 3a). *Caballeronia*, *Paraburkholderia*, and *Burkholderia* are either monotrichous or lophotrichous[16,17], *Pandoraea* are monotrichous[18], whereas *Capriavidus* are peritrichous[19] (Table S1). While all these species swam freely at 15–34 μm s⁻¹ on average in the standard chamber (Fig. 3b), their movements in Q-1D showed a clear difference (Fig. 3c). Eight species, including *C. insecticola* and *Pandoraea norimbergensis*, exhibited smooth and directional movement with a net displacement of 40–130 μm min⁻¹ on average in Q-1D, and 3–10% of their average speed in the standard chamber (Table S1 and Movie S7). In contrast, five species including *B. anthina* were barely able to move in Q-1D with a displacement of 0-6 μm min⁻¹, a decrease to 0-0.4% of their instantaneous speed in the standard chamber (Movie S8). These results demonstrated that ability to move through confined passages varies even in closely related bacterial species carrying polar flagella. Notably, ability to move within confined spaces positively correlated with infection rates of these bacterial species in the bean bug host (Fig. 3d), with bacteria showing smooth and directed movement in Q-1D having an almost 100% infection rate while bacteria with poor movement in Q-1D showing no infection. Three evidence support the relevance of the Q-1D device as a model for the host CR: (i) *C. insecticola* exhibits comparable net displacements in dissected CR-M4B tissue and in the Q-1D channels (Fig. 1e and Fig. 2d), (ii) the ability of closely related species to move efficiently in Q-1D positively correlates with their infection rates of the host (Fig. 3d), and (iii) the addition of methylcellulose in Q-1D reduces particle diffusion in a manner consistent with the mucus-like CR matrix (Fig. S1). Together, these results indicate that the Q-1D device captures essential mechanical and transport constraints relevant to the CR environment.

To clarify why there is such a difference in movement between closely related species with polar flagella, we visualized how they use their flagellar filaments within standard chamber and Q-1D by amine-reactive dye staining[13]. Among the 13 species, we were able to visualize the flagellar filaments of four species, *C. megalochromosomata*, *B. anthina*, *P. norimbergensis* and *P. oxalativorans*, as in *C. insecticola* (Table S1). High-speed imaging revealed that *C. megalochromosomata*, *P. norimbergensis* and *P. oxalativorans* wrapped their flagellar filaments around their cell body during switchbacks under high viscosity (Fig. S6a and Movie S9), as seen in *C. insecticola*[13]. The helical shapes of the flagellar filaments were consistent with normal and coil forms in polymorphic transformations[20], changing from 2 to 1 μm in helix pitch and from 0.3 to 0.5 μm in helix radius after flagellar wrapping (Fig. S6b). However, *B. anthina* exhibited no flagellar wrapping (Fig. S6a and Movie S9); the helical filaments folded longitudinally proximal to the cell surface in an apparent ring with larger diameter than that found in the normal form. Furthermore, flagellar wrapping was observed at a high frequency in *C. insecticola* and *P. norimbergensis* in Q-1D, while no wrapping was observed in *B. anthina*, as in the standard chamber, (Fig. 3e and Movie S10). Notably, in *C. insecticola* and *P. norimbergensis* the ratio of wrapped cells in Q-1D increased to 65%, 3-4 times higher than in the standard chamber (Fig. 3f). These results suggest that these bacterial species, presumably triggered by spatial limitations, change their swimming style to be biased toward flagellar wrapping to achieve efficient locomotion in narrow passages, resulting in directional movement.

To understand how cells reverse their direction of movement within the Q-1D, we first performed high-speed fluorescence imaging of flagellar filaments in *Salmonella*, a peritrichous bacterium. During directional switching, the cell body retained its orientation, while the bundle of flagellar filaments at the rear unbundled and bundled again on the opposite pole, resulting in propulsion to the other direction (Movie S11). Notably, *Salmonella* cells consistently swam with their flagellar bundle trailing behind the cell body, regardless the direction of movement. We speculate that nanoscale gaps between the Q-1D and the glass surface allowed sufficient space for polymorphic transformation of the flagellar filaments. In addition, a small gap may exist

between the cell and the channel wall, allowing partial protrusion of the flagellar filaments. We regard this as a current technical limitation of device fabrication, and improving the precision of the microchannel geometry to minimize such nanoscale gaps will be an important direction for future development. In contrast, in *C. insecticola* (Movie S12), the flagellar filaments undergo a dynamic transition between wrapped and unwrapped configurations. When the filaments were unwrapped, they trail behind the cell, and the cell propelled forward. Upon reversal of motor rotation, the cell moves in the opposite direction, with the flagellar filaments wrapped around the front of the cell body. These findings demonstrate that in narrow passages, polar-flagellated *C. insecticola* exhibit direction-dependent flagellar configuration: they swim with wrapped flagellar filaments in one direction, while in the opposite direction, the flagellar filaments remained unwrapped.

We used numerical simulations to interrogate why flagellar wrapping is dominant in narrow passages. A fluid-mechanical model of a swimming bacterium shows two distinct modes of unwrapped and wrapped flagella mode (Fig. S7a), and the bacterial models were considered to be swimming unidirectionally in a narrow tube (Fig. S7b and Supplementary Note 1). In this study, we employed linear Newtonian and Maxwell fluid models as minimal theoretical frameworks to discuss the swimming efficiency of the wrapping mode in confined environments. In contrast to the flow field in free space (Fig. S7c), we found that in the narrow tube, the unwrapped flagellum does not generate any significant flow from the lateral side of cell body (Fig. 3g Top). The fluid friction between the cell body and the wall provides strong resistance such that the flow only contributes to the agitation of the fluid around the flagellum. On the other hand, the wrapped flagellum scrapes the fluid in the gap like a corkscrew, creating a laminar flow structure in the narrow tube and contributing to cell propulsion (Fig. 3g Bottom, and Fig. S7c). This seems to explain our observations of cells swimming within Q-1D (Fig. 3i and Movie S13). Here, $\Delta R$ represents the radial clearance between the bacterial surface and the channel wall. We defined the gap as normalized $\Delta R$ by ($H/2$), where $H$ is the cell length, to define a dimensionless parameter, facilitating comparison between different bacterial sizes. The speed of the unwrapped mode decreases monotonically as the gap between the wall and the bacterium becomes narrower (Fig. 3h Red). However, the swimming speed of wrapped-mode cells is maintained even in the narrow tube (Fig. 3h Blue). Assuming a cell length of $H = 2.5\,\mu m$, the gap in a circular tube of 1 μm diameter is $\Delta R/(H/2) = 0.025$, and the swimming speed is calculated to be $U = 180\,\mu m/min$ for the wrapped mode, which is comparable to the observed speed of *C. insecticola* (Table S1). The effect of the gap on the rotational angular velocity is relatively small in both modes (Fig. S7d), indicating that the flagellar motor produces sufficient torque to generate cell rotation even in a narrow channel. We modeled the channels as circular tubes for simplicity and numerical stability. Although the experimental microchannels have a rectangular cross-section, we expect the essential hydrodynamic behavior to remain qualitatively similar, as the dominant effect arises from the narrow confinement rather than the cross-sectional shape. We emphasize that our simulations were performed under the working hypothesis of a quiescent (no bulk-flow) CR environment. Under this assumption, our hydrodynamic modeling considers bacterial motility in a viscous fluid; however, we confirmed that the results are qualitatively consistent with those in viscoelastic fluid (Supplementary Note 1). Compared with a recent numerical study[21], our model highlights a distinct regime in which the swimming speed of wrapped mode exceeds that of unwrapped mode in tightly confined spaces. These results confirm that there is a physical advantage of flagellar wrapping in narrow passages, and therefore wrapped-mode swimming is likely dominant in such confined environments.

## Moderately flexible hook for flagellar wrapping

We wondered which flagellar component/feature is responsible for wrapping motility. In *Campylobacter* and *Shewanella*, which exhibit wrapping motility, flagellar filaments are composed of two types of flagellin monomers (FlaA/FlaB), and it is thought that the flagellin comprising the cell-proximal portion of the flagellar filament provides the flexibility for the flagellar filament to wrap around the cell body[22,23]. Since *C. insecticola* has only one flagellin gene (*fliC*)[24], we focused on the hook, the 60 nm long structure that connects the flagellar motor to the filaments[12,25], as a candidate that could contribute to the flexibility of the flagellar filament required for wrapping. When cells were immobilized on a glass slide and flagellar rotation was inactivated by carbonyl cyanide 3-chlorophenylhydrazone (CCCP), the filaments at the cell-proximal end wobbled at a small angle due to Brownian motion (Movie S14). Based on this observation, the thermal fluctuation of fluorescently labeled flagellar filaments was measured as the flagellar orientation angle at 5-ms time resolution (Figure S8), the variances of the orientation angle ($\sigma = 0.038 \pm 0.008$ for *C. insecticola* and $\sigma = 0.018 \pm 0.006$ for *B. anthina*), were used to obtain the hook bending stiffness, $A_{hook}$[12]. Assuming that the number of flagellar filaments is three, the stiffness of *C. insecticola* ($A_{hook} = 0.67 \times 10^{-25}$ Nm$^2$) is 5 times lower than that of *B. anthina* ($A_{hook} = 0.15 \times 10^{-25}$ Nm$^2$) (Supplementary Note 2), consistent with the previously reported value measured under a low torque regime in the flagellar motor of *Escherichia coli*[25].

Next, we constructed a mechanical model of flagellar wrapping, in which the flagella bundle is regarded as a single elastic helical filament with short-range attractive interactions (Fig. S9a and Supplementary Note 2). For the deformations of a flexible filament, we computed all the elastic modes of stretching, bending, and twisting, taking full account of the geometric nonlinearities inherent in such slender structures, based on a previously reported method[26]. Three qualitatively distinct wrapping behaviors were observed in simulations with varying hook/filament stiffness ratios, $A_{hook}/A$, and filament twist/bend stiffness ratios, $C/A$ (Table S2 and Fig. S9b). A flagellar filament with a flexible hook at $A_{hook}/A = 0.02$ and $C/A = 0.75$ showed a smooth transition to wrapping around the cell body (Movie S13), as observed in *C. insecticola* (Fig. 4a Left). On the other hand, a filament with a more rigid hook at $A_{hook}/A = 0.14$ and $C/A = 0.75$ performed incomplete wrapping that overlapped near the end of the cell body (Movie S16). For an even smaller twist/bend stiffness ratio at $C/A = 0.5$, this feature was more evident, where the filament folded into a ring at the cell-proximal end without wrapping at all (Movie S17), as we observed in *B. anthina* (Fig. 4a Right, and see also Movie S18). Our mechanical modeling and numerical simulations suggest that flagellar wrapping can be explained by a single factor, the stiffness of the hook in polar flagellated bacteria. Previous modeling studies have demonstrated that both hook stiffness and motor torque play key roles in flagellar wrapping[27]. While torque effects are not addressed in the present study, we are currently investigating this factor in a separate work[28]. Our simulations successfully reproduce diverse morphology of flagellar filaments observed in the experiments, thus providing complementary insights to those of the previous study[27].

To test this model, we constructed hook-swapping mutants by expressing the hook component protein FlgE of *B. anthina* (FlgE$_{Ba}$) in *C. insecticola* and FlgE of *C. insecticola* (FlgE$_{Ci}$) in *B. anthina* (Fig. 4bc). The swimming speed of both swapping mutants was almost the same as that of their wildtype (Fig. 4d Left), while the $\sigma$ was measured to be 0.011 for *C. insecticola* with FlgE$_{Ba}$ and 0.051 for *B. anthina* with FlgE$_{Ci}$ (Fig. 4d Center). We found *C. insecticola* with FlgE$_{Ba}$ decreased cell displacement in Q-1D to almost zero, whereas *B. anthina* with FlgE$_{Ci}$ increased its cell displacement in Q-1D but did not reach the level of wildtype *C. insecticola* (Fig. 4d Right). Indeed, *B. anthina* with FlgE$_{Ci}$

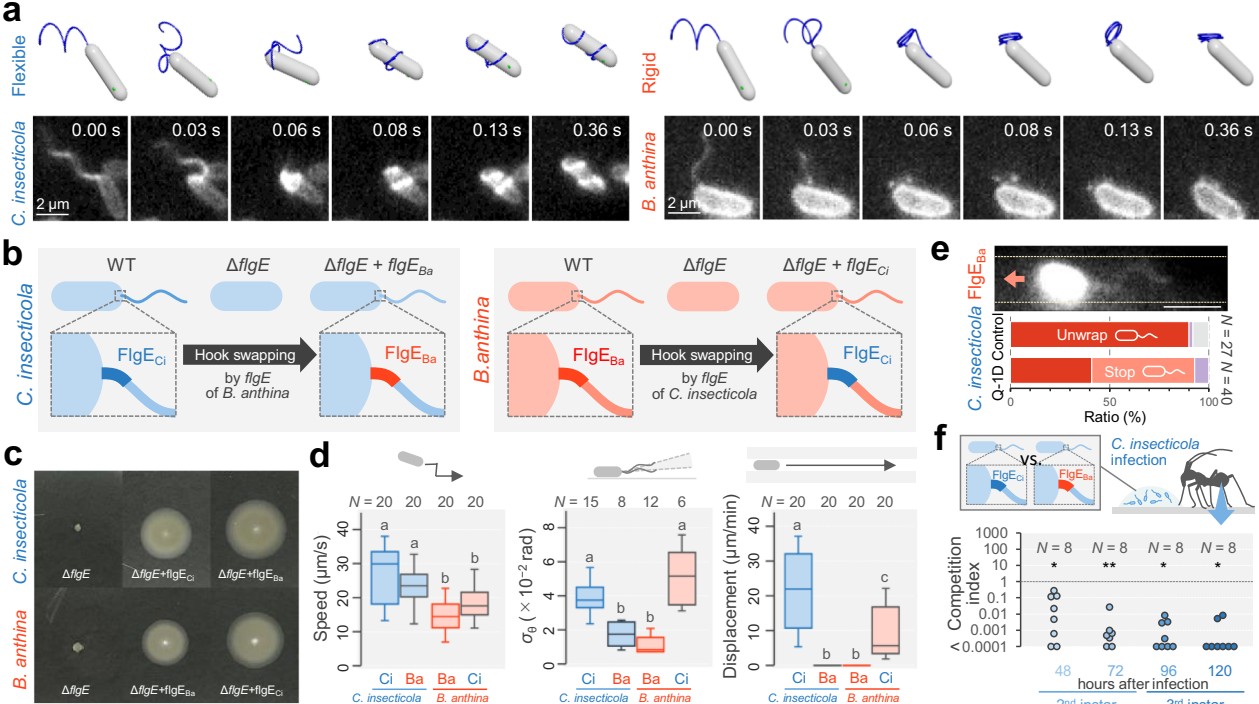

**Fig. 4 | Moderately flexible hook for flagellar wrapping. a** Sequential images of the flagellar filament during cell switchbacks. Upper: Numerical calculation of the flexible and rigid hook. The coil form of flagellar filaments is transformed by the CW rotation. Lower: Fluorescent images of *C. insecticola* and *B. anthina*. See also Movie S18. **b** Schematic of the genetic swapping of hook by *flgE* deletion and transformation. **c** Motility assay on soft agar of the *flgE* swapping mutants. **d** Characterization of *flgE* swapping mutants. Left: Swimming speed for 1.0 s in growth medium. Center: Variance of the flagellar orientation angle $\sigma$. Right: Cell displacement for 1 min in Q-1D. Box plots present the minimum, maximum, sample median, and the first and third quartiles. Schematic of the measurements is presented at the top of each graph. Different letters indicate statistically significant differences (two-sided Wilcoxon rank-sum test with Bonferroni correction, $p < 0.05$; exact $p$ values and test statistic are provided in Source data). **e** Direct visualization of flagellar filaments in the *flgE* swapping mutants. Top: Fluorescent

image in Q-1D. The edge of the narrow passages is represented by dashed lines. The arrows indicate the direction of movement of the cell. Bottom: Fraction of flagellar filament morphology in growth medium and Q-1D. The color codes of bars are the same as in Fig. 3f. **f** In vivo competition assay. Infection competitiveness of *C. insecticola* with FlgE$_{Ci}$ against *C. insecticola* with FlgE$_{Ba}$ in the gut symbiotic organ of *R. pedestris*. The competitive index (CI) was calculated by (output *C. insecticola* with FlgE$_{Ba}$ colony count/input *C. insecticola* with FlgE$_{Ba}$ colony count)/(output C. *insecticola* with FlgE$_{Ci}$ colony count/input *C. insecticola* with FlgE$_{Ci}$ colony count). CI represents the relative colonization success of *C. insecticola* strains in the gut symbiotic organ. A lower CI indicates that C. *insecticola* with FlgE$_{Ci}$ outcompeted *C. insecticola* with FlgE$_{Ba}$. Two-sided one-sample Wilcoxon signed-rank tests (against CI = 1) revealed significant differences from CI = 1 at 48, 72, 96, and 120 h ($V = 0$; $p = 0.014$, $p = 0.008$, 0.013, and $p = 0.010$, respectively). Source data are provided with this paper.

started to perform flagellar wrapping (Movie S19), although infrequently, potentially due to the swapped hook being too flexible to fully replicate the behavior of wildtype *C. insecticola*. In contrast, *C. insecticola* with FlgE$_{Ba}$ was no longer able to wrap their flagellar filaments around their cell bodies in Q-1D (Fig. 4e) and showed a strong reduction of their infection competitiveness in the host *R. pedestris* (Fig. 4f). Together, these results demonstrate a pivotal role of the hook not only in the flagellar wrapping but also in cell displacement in Q-1D and in insect-microbe symbiosis.

## Discussion

We investigated the physical and ecological role of flagellar wrapping in insect symbionts, revealing that this mode of flagellar motility gives bacteria an advantage in confined spaces (Fig. 2 and Fig. 3) and promotes the establishment of symbiosis (Fig. 1 and Fig. 4). Importantly, whereas the phenomenon of flagellar wrapping in the symbionts of *C. insecticola* has been described previously[13], to our knowledge no previous study has directly demonstrated that wrapping confers a locomotory advantage specifically within micrometer-scale constrictions. By combining in vivo imaging, Q-1D microfluidics, hydrodynamic modeling and genetic modification, our results are the first to demonstrate that confinement itself promotes wrapped configurations and that these configurations mechanically enhance translocation in one-micrometer-scale passages. Although

simplified, the Q-1D device reproduces key physical features of the CR, including geometric confinement, reduced effective diffusion, and comparable cell displacement, and thus provides an appropriate platform for examining how flagellar wrapping affects propulsion under host-relevant mechanical constraints. In addition to insect symbionts and other symbionts such as *V. fischeri*[13,29], serious human pathogens such as *Campylobacter jejuni*, *Helicobacter* spp., and *Pseudomonas aeruginosa* are also known to exhibit flagellar wrapping[23,30,31]. Therefore, it is probable that flagellar wrapping is a common strategy for infecting host animals when viscous conditions, such as mucous and confined regions like gut lumen crypts, are encountered. Moreover, this motility could also be beneficial in complex constricted spaces such as soil, opening a new window to understand the ecology of environmental bacteria.

Although we were not able to visualize the flagella of all tested bacteria, considering that the directional motility in Q-1D is due to flagellar wrapping (Fig. 2), this characteristic swimming style may be common in *Burkholderia* sensu lato. The flagellar wrapping in *Pandoraea*, the sister group of *Burkholderia* sensu lato, suggests that this motility was developed by the common ancestor of these bacterial groups, and *B. anthina* reverted to an unwrapped swimming style (Fig. 2 and Fig. S6). From an evolutionary perspective, it would be of great interest to understand the selective forces that have led some species to wrap while other closely related species do not. This study

demonstrated that hook flexibility is important for flagellar wrapping; however, this flexibility may potentially reduce rotation efficiency in the unwrapped mode due to reduced torque transmission. Although our experimental data do not directly measure this effect, it is plausible mechanical trade-off that warrants further study. If so, there may be a trade-off between the wrapped and unwrapped motility and, in some environmental niches, flagellar wrapping and hook flexibility may be disadvantageous.

Another notable point is that wrapping motility could be induced by a mechanical stimulus in narrow passages (Fig. 3 and Fig. 4). In *V. fisheri*, cells alter their behavior upon entry into a confined space, allowing them to escape from confinement[32]. In *Shewanella putrefaciens*, flagellar wrapping enables efficient movement away from a substratum using instantaneous CW rotation of the flagellar motor[14,15]. Similarly, in *E. coli*, CW bias gradually increases under high load[33]. Interestingly, the observed directional persistence may reflect a physiological response of the bacteria to spatial confinement. As proposed in previous studies[34], wrapped-mode swimming has been associated with chemotactic behavior. In our case, the directional bias could indicate a chemotactic-like response, or alternatively, an intrinsic motility pattern that favors directional movement as a strategy to escape highly confined regions where chemical diffusion is extremely limited. Further studies will be needed to clarify the underlying mechanisms. Wrapped-mode bias in narrow passages may be explained by a universal physical process such as the loads triggered by spatial limitations and intrinsic properties of the flagellum itself. Alternatively, wrapped-mode bias could be due to an unknown physiological pathway in bacteria that senses ambient physical conditions.

Since the rotational model of the bacterial flagella was established in 1974[35-37], the machinery and mechanisms of flagella have been widely investigated, but flagellar wrapping is a much more recent discovery. Although the number of culturable bacteria is still extremely limited, a wide variety of movement patterns and machinery have been found in bacteria[5], but their ecological significance and evolutionary process are still largely unknown[38]. Mimicking their microhabitat in microfluidic devices in conjunction with analyzing the physical properties and genetic backgrounds of different species, as shown in this study, will reveal new bacterial behaviors and adaptive strategies in natural environments.

## Methods

### Strain and culture conditions

*C. insecticola* (former called *Burkholderia insecticola*) RPE64 cells and other bacteria belongs to *Burkholderia sensu lato* group were grown to an early log phase in yeast-glucose (YG) liquid medium or agar plate [0.5% (wt/vol) yeast extract, 0.4% (wt/vol) glucose and 0.1% (wt/vol) NaCl] at 28 °C[7]. *S. enterica* was grown in LB liquid medium [1% (wt/vol) tryptone, 0.5% (wt/vol) yeast extract and 0.1% (wt/vol) NaCl] at 37 °C[39]. *V. fisheri* was grown to an early log phase in seawater tryptone (SWT) liquid medium [0.5% (wt/vol) tryptone, 0.3% (wt/vol) yeast extract, 0.3% (wt/vol) glycerol, 70% (vol/vol) artificial seawater] at 28 °C[40]. Bacterial species used in this study and their culture conditions are listed in Table S3.

### Construction of deletion mutants and complemented strains

Flagella hook gene, *flgE* (BRPE64_ACDS27220 of *C. insecticola* or BAN20980_05336 of *B. anthina*), was deleted by the homologous recombination-based method with the suicide vector pK18mobsacB[41]. The successful deletion was confirmed by Sanger sequencing. Gene complementation was performed with the broad-host-range vector pBBR122. The fragment of *flgE* gene and its upstream region from *C. insecticola* or *B. anthina* was cloned into the Nco1 site of pBBR122 with NEBuilder HiFi DNA Assembly Master Mix (New England Biolabs). The constructed vectors were then introduced into the deletion mutants

by electroporation. Primers used in this study were listed in Table S4. Strains and mutants used in this study were listed in Table S5.

### Microfluidic device

Microfluidic devices (Q-1D) were fabricated using standard photolithography and soft lithography methods[42]. Briefly, polydimethylsiloxane (PDMS, Sylgard 184, Dow), a two-part silicone elastomer, was cast over a photolithography master and cured at room temperature for 48 h. The photolithography master was prepared by a photoresist (OFPR 800 23cp, Tokyo Ohka, Japan) pattern on a flat silicon wafer beforehand. The photoresist was spin-coated with a speed of 4000 rpm for 60 s to have ~1 μm thickness, and photolithographically patterned to have 1.5 μm wide stripes. The PDMS casting and curing on the master replicates the stripe patterns, making microfluidic channels with similar dimensions with the master pattern in a negative/positive inverted manner. A piece of PDMS was cut out using a scalpel and used as a microfluidic device. Cell suspensions with YG medium containing 0.4–0.5% methylcellulose (M0512, viscosity: 4000 cP at 2%, Sigma-Aldrich) were dropped onto a glass slide and then covered with the microfluidic device casting from the top, which traps bacteria in the channels. Bacterial cells were not introduced from the open ends of the channels. Instead, a droplet of cell suspension was placed between the PDMS device and the glass coverslip, allowing cells to become passively confined within the Q-1D channels during assembly of the chamber. The cell behaviour was observed within 20 min after the confinement. Fluorescent beads (Cat No. F8811; Thermo Fisher) were diluted 1/200 in water containing 2% bovine serum albumin and 0.4% methylcellurose.

### Phylogenetic tree

Genome data of representative species of the genus *Burkholderia sensu lato* and related beta-proteobacteria were downloaded from the NCBI Assembly database. The multiple alignments of the 16S rRNA gene were constructed using Clustal W and displayed by iTOL (https://itol.embl.de/).

### Measurements of swimming speed

The cell culture was centrifuged at $10,000 \times g$ for 4 min, and the pellet was suspended in fresh liquid medium. The sample chamber was assembled using a coverslip with two pieces of double-sided tapes to create a thin channel for observing bacterial swimming motility. The cell suspension was poured into the chamber, and both ends of the chamber were sealed with nail polish to keep the sample from drying.

### Preparation of fluorescently labeled cell

Cell culture of 250 μL was centrifuged at $10,000 \times g$ for 1 min at 25 °C, and resuspended in 1 mL of 0.1 M sodium phosphate buffer (pH 7.5). The suspension was mixed with 25 μg of DyLight 488 NHS ester (Thermo Fisher) or Cy3 NHS ester (Lumiprobe) and incubated at room temperature for 10 min. The suspension was washed twice in the sodium phosphate buffer, and resuspended in the motility buffer (20 mM potassium phosphate buffer pH 6.0 and 20 mM glucose).

### Measurement of hook flexibility

The surface of the coverslip for sample observation was pre-treated with collodion to immobilize the cell on the surface. The fluorescently-labeled cells suspended in phosphate-buffered saline (PBS), made with 75 mM sodium phosphate (pH 7.4) and 68 mM NaCl in the presence of 50 μM CCCP at the final concentration were poured into the chamber, and both ends were sealed with nail polish to keep the sample from drying. The bacteria that tightly attached to the glass surface were used for observation in fluorescent microscopy. The hook bending stiffness, $EI$, was estimated using the formula $EI = k_B T L_{hook}/\sigma^2$, where $k_B$ is Boltzmann's constant, $T$ is the absolute temperature, a length of the hook $L_{hook}$ (assumed to be 60 nm), and $\sigma$ is the variances of the

orientation angle of the flagellar filaments[12,25]. Assuming that the flagellar bundle in *C. insecticola* and *B. anthina* consists of three filaments, we defined the stiffness per hook as $A_{hook} = EI/3$.

## Optical microscopy

Bacterial movements in Q-1D and the chamber were visualized under a phase-contrast microscope (IX73, Olympus) equipped with a 20× objective lens (UCPLFLN20X, NA 0.70, Olympus), a CMOS camera (DMK 33UX174, Imaging Source, or Zyla 4.2, Andor), and an optical table (HAX-0806, JVI). Projections of the images were captured as greyscale images with the camera under 0.1-s resolution and converted into a sequential TIF file without any compression. All data was analyzed by ImageJ 1.53k (rsb.info.nih.gov/ij/) and its plugins, TrackMate[43], and particle tracker[44].

For direct visualization of fluorescently labeled cells in both the Q-1D and the host midgut, the sample was examined under an inverted microscope equipped with a ×100 objective lens (UPLXAPO100×OPH, NA1.45, Olympus), a dichroic mirror (Di02-R488, or Di02-R532, Semrock), and an emission filter (FF01-540/80, Semrock, or FELH0550, Thorlabs). A laser beam (488-nm or 532-nm wavelength, OBIS488 or OBIS532, Coherent) was introduced into the inverted microscope through a lens for epi-fluorescence microscopy. Projections of the images were captured with the camera under a time resolution of 5–10 ms.

To observe the symbionts in real-time within the host midgut, the bacterial cells in the midgut were observed under an inverted microscope equipped with a ×40 objective (LUCPLFN40X, NA 0.60, Olympus), a filter-set (59022, Chroma), and a mercury lamp (U-HGLGPS, Olympus). Projections of the images were captured with the camera under 0.2-s resolution.

## Numerical calculation

Visualizations of simulated configurations are made using a free ray-tracing software POV-Ray3.7.0 (Unofficial version for Intel Mac OS). A movie file is produced from a series of configuration images by the command FFmpeg.

## Motility assay on soft agar plate

Colony of *C. insecticola* and *B. anthina* were spotted on soft agar plate containing 0.4% agar and YG medium. For preparation and cultivation of the mutant strains, antibiotics were added to the medium at the following concentrations: kanamycin 30 and 100 $\mu$g ml$^{-1}$ for *C. insecticola* and *B. anthina*, and chloramphenicol 30 $\mu$g ml$^{-1}$ for *B. anthina*. Spreading of bacteria on the plate was captured after 24 h of incubation at 28 °C.

## Insects rearing and symbiont infection

The bean bug *R. pedestris* was reared in the laboratory under a long-day regimen (16 h light, 8 h dark) at 25 °C in petri dishes (90 mm in diameter, 20 mm high)[10]. Soybean seeds and ion exchanged water were provided to the bugs as food and water, respectively. Newly molted second instar nymphs were fasted from water overnight, and then fed with suspension of symbiotic bacteria expressing mCherry or GFP[45] or labeled cell body and flagellar filaments by DyLight 488 NHS ester (Thermo Fisher) (see the section of Preparation of fluorescently labeled cell). The symbiotic organs of the nymphs were dissected in 10 mM PBS buffer within 2–4 h after feeding. The samples were gently placed on a coverslip, and the sample chamber was assembled with two coverslips and two pieces of double-sided tape to act as a thin channel for observation under an optical microscopy.

## Competitive infection assay

A bacterial solution containing each 5000 cells/$\mu$l of [*C. insecticola* $\Delta flgE + flgE_{Ci}$] ( = Native flagella symbiont) and [*C. insecticola* $\Delta flgE +$ $flgE_{Ba}$] ( = Swapped flagella symbiont) were mixed, and 1 $\mu$l of the bacterial mixture was fed to the freshly molted second instar nymphs of the bean bug. The gut symbiotic organ (crypt-bearing M4 region) was dissected 48, 72, 96, and 120 h after inoculation, homogenized by a pestle, and total DNA was extracted by QIAamp DNA Mini Kit (Qiagen) and subjected to qPCR to determine bacterial titers. Due to the lack of effective antibiotic resistance markers, we decided to estimate bacterial titers in the symbiotic organ by qPCR specifically targeting $flgE_{Ci}$ and $flgE_{Ba}$[46]. qPCR was conducted with specific primer sets for $flgE_{Ci}$ (F[5′-TATCAGCTGTCGAACAACGG-3′] and R[5′-ATGTTGCCGTTGTCATCGAG-3′]) and $flgE_{Ba}$ (F[5′- CAGA TTTCTCGAAACGGAGACC-3′] and R[5′- ATCGAGTTCGCGTACATG TC-3′]). By using the bacterial titers, competitive index (CI) values were calculated by (output Swapped flagella symbiont/input Swapped flagella symbiont)/(output Native flagella symbiont/input Native flagella symbiont) and statistically evaluated by the 1-sample $t$ test (against CI = 1.0).

## Statistical analysis

All statistical analyses were performed by GraphPad Prism 9.1.0 (GraphPad Software) and R ver 4.1.3[47].

## Reporting summary

Further information on research design is available in the Nature Portfolio Reporting Summary linked to this article.

## Data availability

All data generated in this study are provided in the Supplementary Information/Source data file with this paper. Source data are provided with this paper.

## Code availability

All data are available in the main text, the Supplementary Information/ Source data file with this paper, or Zendo (https://doi.org/10.5281/zenodo.17646174).

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

## Acknowledgements

We thank Eli Cohen, Daisuke Takagi, Jonathan Lynch, and Peter Mergaert for critical reading and comments on the manuscript. This study was supported partly by KAKENHI grants from the Japan Society for the Promotion of Science (JSPS) 22H05066 to D.N. and T.K., 22H05067 to H.W. and 22H05068 to Y.K. and K.T., FOREST Program from Japan Science and Technology Agency (JST) JPMJFR2411 to D.N., and funds from Noguchi Institute and Precise Measurement Technology Promotion. The funders had no role in study design, data collection and analysis, decision to publish, or preparation of the manuscript.

## Author contributions

Conceptualization: H.W., T.K., Y.K., D.N., Methodology: All authors, Formal analysis and investigation: All authors, Funding acquisition: H.W., Y.K., D.N., Writing—original draft: A.Y., W.H., Y.K., D.N., Writing—review & editing: Y.K., D.N.

## Competing interests

The authors declare no competing interests.
