## [Transparent Peer Review file · Nature Communications]

Bacteria break through one-micrometer-square passages by flagellar wrapping

Corresponding Author: Professor Daisuke Nakane

Version 0:

Reviewer comments:

Reviewer #1

(Remarks to the Author)

In the manuscript by Yoshioka et al., the authors demonstrated an interesting physiological function of a recently reported mode of flagellar motility, the wrapping mode. Through direct imaging, they discovered that the stinkbug symbiont *C. insecticola* (formerly known as *Burkholderia* sp. RPE64) employs the wrapping mode of motion to efficiently navigate the approximately 1-micron-diameter gut passage and reach the stinkbug's symbiotic organ. The authors utilized microfluidic experiments and hydrodynamic modeling to show that flagellar wrapping enables the cell to control fluid flow and generate propulsion for movement in narrow passages. They further employed physical simulations and gene-swapping experiments to reveal that moderate hook flexibility is essential for wrapping motility.

The potential role of wrapping motility in *C. insecticola* colonizing the stinkbug's symbiotic organ was suggested in Ref. 13. Now, the authors in this manuscript have advanced this understanding by demonstrating this essential role under physiological conditions, specifically within the CR tube of the bug. This study provides an excellent example of the biological and ecological function of wrapping motility. Moreover, the authors elucidated the detailed biophysical mechanisms of the phenomenon. Overall, the study is of suitable quality for Nature Communications.

However, before recommending publication, I hope the authors can clarify the following questions:

1. For the major experiment in Fig. 1, the motion of *C. insecticola* cells was observed in dissected symbiotic organs. Would this dissection affect the motion pattern of cells and alter the conclusions? For instance, was there flow in the CR, and if so, how would the flow affect the motion of bacterial cells? The authors could use tracers to examine the flow in the CR.
2. What accounts for the direction bias of cells towards the symbiont organ? In the Q-1D experiments, cells move persistently along either direction with wrapping motility, but within the insect body, cells must navigate towards the symbiont organ. Is there an asymmetry of the CR that could account for the direction bias?
3. In the Q-1D experiments, the authors use viscous solutions of 0.4% methylcellulose (MC). Is this solution a good model of CR mucus fluid? Measuring the viscoelasticity of CR fluid and comparing it with that of the MC solution would be helpful.
4. The authors' hydrodynamic modeling considers bacterial motion in a viscous fluid. However, CR mucus fluid and the MC solution used in the Q-1D experiments are both viscoelastic. Viscoelasticity of fluids is known to alter bacterial swimming behavior near solid surfaces. I wonder if any results would be modified if viscoelasticity is taken into account. Also, there is a recent numerical study on the propulsion of cells with wrapping motility (Gidituri et al, JFM 2024). Could the authors discuss the differences in their model?

Minor points:

Line 18: Why would limited diffusion impair the movement of cells? Is the diffusion here referring to active diffusion?

Line 88: What is the relation between the *cheA* mutation and clockwise rotation of the motor?

Line 174: Hook bending stiffness was estimated by the thermal fluctuation of flagellar filaments. The authors should provide the details of the calculation or theory behind this important measurement.

Reviewer #2

(Remarks to the Author)

In this manuscript, the authors report experimental results, supported by numerical modeling work, that demonstrate a key role of flagellar wrapping in navigating narrow channel structures as they are often encountered by motile bacteria. They mostly focus on the specific example of the stinkbug symbiont *Caballeronia insecticola* and present motility data both from the host's gut sorting organ and from PDMS-based microfluidic chips. They also extend their analysis to more than 10 related species and identify the mechanical properties of the hook as the key determinant of flagellar wrapping.

The manuscript is based on thorough experimental data, in particular the large numbers of cells that can be tracked in the linear microfluidic channels is a nice feature of this work, while the fluorescence imaging data seems a bit less abundant. Also the attempt to connect fundamental physical properties of the bacterial flagellum to biologically relevant questions related to insect symbionts are a nice feature of this work. However, several aspects remain unclear and require revision before publication in *Nature Communications* can be considered.

Major points:

- At the beginning of the section on "Flagellar wrapping gives an advantage in narrow passages" you show results on *Salmonella* in narrow channels. This is an interesting addition because of *Salmonella*'s peritrichous flagellation compared to the polar flagellation of the other species discussed. However, the *Salmonella* part remains somewhat incomplete and unsatisfactory because no fluorescence data is included that shows how *Salmonella* actually operates their flagella to move in the channels. Do they switch from push to pull upon a change in direction? Or do they somehow reassemble their bundle on the other side of the cell body to reverse their direction of motion? This would be very interesting to know, as peritrichous bacteria normally do not move in pulling mode in open liquid. Why is no such data included? If labeling of the flagella failed for *Salmonella*, the authors could have chosen *E. coli* as the standard example of a peritrichous swimmer instead. I suggest to include such data.

- Fig. 2c: I am confused about the trajectories shown in panel 2c. For a polarly flagellated swimmer, I would expect repeated reversals in swimming direction when the motor switches from CCW to CW and back (as for example in *V. alginolyticus*, *P. putida* and others). But here the trajectories look smooth, i.e. without reversals or other abrupt changes in swimming direction. Why are there no turns? In general: what does the swimming pattern of *C. insecticola* look like in open bulk liquid without any confinement? Is there no literature on this? I would expect that this is investigated before moving to confined environments.

- I am very confused about the directional bias reported in Fig. 2d. How do you define "forward" and "backward" directions? If your experimental setup (microfluidic device, distribution of cells, nutrients etc.) is symmetric, there is no reason why one of the directions along the microfluidic channel should be preferred, i.e., cells should, on average, move left or right with the same probability. Perhaps you have inserted cells only from one side, so that there is a gradient in medium concentration or quorum sensing factors or other chemical cues? From my understanding, the directional bias may be the signature of a chemotactic response (see, for example, Alirezaeizanjani et al, *Sci. Adv.* 2020; 6:eaa6153, where the wrapped mode was identified as the chemotactically relevant swimming mode). Another possibility would be that all cells are entering the channels from one side with the motor ahead, so that there is a systematic difference between "forward" and "backward"? This aspect is not addressed in the manuscript, not even in a speculative manner. You have to address and explain (or at least offer a plausible explanation) for the observation of the bias.

- Along the same lines: You show that cells move through the channels in wrapped mode. But what happens when they reverse their direction of motion? Do they stay in wrapped mode but simply move backward? Or do they unwrap their flagella and switch to push mode? Since you show movies with fluorescently labeled flagella in the channel, this information must be available. Please show fluorescence movies, where cells can be seen that switch their direction of motion inside the channel.

- It would be helpful to be more explicit about the flagellation of the species you investigate. The statement on line 106 "To further investigate the importance of polar flagella..." suggests that they are all polarly flagellated. Is this true? And if yes, are they monotrichous or lophotrichous? Wrapping has been demonstrated for both cases but there may be still differences, see Park et al., *Phys. Fluids* 36, 101917 (2024).

- The terminology (directional versus random) in the discussion of the MSD curves (Fig. S4b) is somewhat unclear/misleading. A parabolic MSD curve is a signature of ballistic motion ($x = v t$ and thus $x^2 = v^2 t^2$). But ballistic motion is not necessarily directional. If cells move ballistically in the channel, but with equal probability to the left and to the right, then the MSD will still be parabolic but the average displacement will remain zero, i.e. without directional bias. Similarly, from a linear in time MSD, you cannot conclude that there is no directional bias. To check whether a directional bias is present or not, I suggest that you also compute the mean displacement (not only the mean squared displacement), i.e. that displacement between $t=0$ and $t=t_{\text{end}}$ for each cell, averaged over all cells. This quantity will be non-zero if there is a directional bias.

- For those bacterial strains that do not move within the confined quasi-1D channel environment, such as the *cheA* deletion mutant or some of the related species that were additionally investigated, how were they introduced into the channels in the first place? In your movies, I can see that they are immobile but distributed all across the channels.

- The modeling results showing that the hook stiffness is a key parameter for filament wrapping are not new and have been reported before, see Park et al. Scientific Reports 12, 6482 (2022), <https://doi.org/10.1038/s41598-022-09823-4>. Park et al. furthermore showed that not only the hook stiffness but, in addition, also the torque of the flagellar motor determines whether transitions to the wrapped mode occur, which is not even addressed here. The work of Park et al. has to be included and discussed here. Currently, I do not see any new insights from the hook modeling reported here that goes beyond what is already known from Park et al.

Smaller points:

- page 3: "...whereas the forward directional bias remained relatively consistent over time..." What do you mean by "consistent"?

- page 5: "...two distinct models" maybe you mean "modes" instead of "models"?

- page 5: "...no lateral suction flow..." How could there be such a lateral flow if there are the channel walls? How would it show up if there was such a flow?

- page 5: What is ΔR ? Why do I need the cell length in this calculation? It is not clear what exactly is calculated here, please explain.

- Related to this: in the experiments you have rectangular channels but in your calculations, if I understand correctly, they are assumed to have a circular cross section. Does this make any difference? Please comment.

- page 5: "estimated" In what sense is this an estimate? Isn't this rather the outcome of a numerical simulation?

- page 7: "inefficient rotation in the normal unwrapped mode" Where can I see this in the reported data?

- Fig. 1g: Is there a way to improve quality of these panels? They are very difficult to read.

- Fig. 2f and Fig. S3d: How was this plot generated? Is there enough data to bin it in intervals of 0.5 sec and still have enough data in each time bin to reliably measure a mean run time? What is meant by "The ratio was sorted by 0.5s..."?

- Fig. 3a: Where do I see the phylogenetic tree of *S. enterica* that you refer to in the caption?

- Fig. 3f: What are the grey parts in the horizontal bar plots?

- Fig. 3h: How was the gap width systematically changed? Can you control the width of the channel with such precision? I could not find anything about this in the methods section.

- Fig. 4f: I am not sure I understand the "competition index", maybe rephrase this?

- Fig. S3d: There seems to be a maximum forward bias at intermediate times. Any idea how to explain this or what this implies?

Version 1:

Reviewer comments:

Reviewer #1

(Remarks to the Author)

I appreciate the authors' effort in addressing the comments. I still have the following concerns:

1. "However, as shown in our previous study (Ref. 3), the constricted region (CR) is a highly narrow passage that is filled with mucin-like mucus and does not allow the passage of liquids such as colored water. This indicates that even under normal physiological conditions, the CR does not support bulk flow. In our current study, we carefully examined the dissected midgut tissues under microscopy and did not observe any visible flow that could influence bacterial motion."

I am not fully convinced that there is no fluid flow in CR after checking Fig. 2 and Fig. 3 of Ref. 3. As the authors noted, the dye stained M1, M2 and M3 but never appeared in the M4B and the M4. If there is no flow in CR, the dye molecules will diffuse into the CR and develop a concentration gradient. However, Fig. 3A of Ref 3 showed a sharp concentration drop at the entry of the CR, which is highly unlikely if the molecules are diffusible and if there is no flow. One explanation for this fact is that there is flow (from M4 to M3 via the CR) flushing the molecules out of CR. Can the authors rule out this possibility? This question is critical because if there is fluid flow in CR, it would change the physical picture of *C. insecticola* navigation process, and the Q-1D microfluidic experiments as well as the hydrodynamic modeling would have to be modified.

2. "To our knowledge, the viscoelastic properties of the CR fluid in *R. pedestris* have not been directly measured. This is primarily due to the extremely small size of second-instar nymphs, only a few millimeters in length, making it technically challenging to obtain sufficient amounts of CR fluid for rheological analysis."

Is it possible to perform microrheology measurement by feeding the bug with microspheres? The measurement can be done at M1, M2 or M3 since the fluid in these regions must be similar as that in the CR.

3. I appreciate the authors' effort to perform additional numerical calculation taking account for viscoelasticity. But 0.4-0.5% methylcellulose solutions also display pronounced shear thinning viscosity. Instead of the Maxwell model, Giesekus fluid model would be more suitable. Anyhow, the calculation is based on the assumption that there is no fluid flow in the CR. If there is flow, the simulation setting would be substantially different.

My overall impression is that the current results in the manuscript have not yet demonstrated the biological and ecological functions of wrapping motility under physiological conditions, specifically within the CR tube of the bug. For this purpose one would have to first characterize the relevant physiological conditions, such as flow and rheology of CR fluids, and use Q-1D device to investigate the role of wrapping motility under those conditions. The current work with Q-1D microfluidic channels has its own right, but that seems more like a sequel of Ref. 13. The channel geometry of Q-1D experiment is relevant, but it is already well known that wrapping mode is beneficial for bacterial navigation in confined space.

Reviewer #2

(Remarks to the Author)

I thank the authors for their thorough revision. All my questions and concerns have been addressed in great detail.

Overall, I am very happy with your revision, it is a beautiful paper and I can now recommend publication in Nature Communications.

Two points remain that I recommend to be addressed:

Following my request, you have added a new movie (S11) that shows a fluorescence microscopy recording of peritrichously flagellated *Salmonella* performing directional reversals in a channel. In the accompanying text, you say that you "speculate that nanoscale gaps between the Q-1D and the glass surface allowed sufficient space for polymorphic transformation of the flagellar filaments." However, in the movie I can see that the flagellar bundle disassembles and spreads widely around the cell body before reassembling on the other side to drive motion in reverse direction. It seems that there is ample space of several micrometers around the cell body, not just "nanoscale gaps". It is a beautiful recording but is this really a channel of 1 μm in width and depth? It seems much larger. I suggest, that you mark the channel walls in your movie (as they are not fluorescent, they are invisible in this recording). An if this is indeed a much wider channel, I suggest that you add more data that actually shows reversals of *Salmonella* in a 1 μm squared channel.

Following another of my requests, you have added Fig. S5c showing the mean displacements. Am I correct in assuming that all these displacements were taken positive? So you actually show the average of the absolute value of the displacement $\langle |x| \rangle$, correct? This is simply the square root of the curves in S5b and does not add any new insight. What I actually had in mind was to show the average displacement taking its sign into account (displacements to the right with a positive sign and displacements to the left with a negative sign). If everything is isotropic, then these curves should be centered around zero (provided you have enough data). This would be a nice test to show that indeed everything is isotropic and there is no chemotactic or other gradient biasing. It would be a nice control and since you have all the data, it should be easy to compute.

I do not need to see the manuscript again and leave it to the editors to check that these points are fixed.

Reviewer #1 (Remarks to the Author):

*In the manuscript by Yoshioka et al., the authors demonstrated an interesting physiological function of a recently reported mode of flagellar motility, the wrapping mode. Through direct imaging, they discovered that the stinkbug symbiont *C. insecticola* (formerly known as *Burkholderia* sp. RPE64) employs the wrapping mode of motion to efficiently navigate the approximately 1-micron-diameter gut passage and reach the stinkbug's symbiotic organ. The authors utilized microfluidic experiments and hydrodynamic modeling to show that flagellar wrapping enables the cell to control fluid flow and generate propulsion for movement in narrow passages. They further employed physical simulations and gene-swapping experiments to reveal that moderate hook flexibility is essential for wrapping motility.*

*The potential role of wrapping motility in *C. insecticola* colonizing the stinkbug's symbiotic organ was suggested in Ref. 13. Now, the authors in this manuscript have advanced this understanding by demonstrating this essential role under physiological conditions, specifically within the CR tube of the bug. This study provides an excellent example of the biological and ecological function of wrapping motility. Moreover, the authors elucidated the detailed biophysical mechanisms of the phenomenon. Overall, the study is of suitable quality for *Nature Communications*.*

However, before recommending publication, I hope the authors can clarify the following questions:

We would like to express our sincere appreciation for the thorough and insightful review. We are grateful for the positive evaluation of our work and for highlighting its contribution to understanding the biological function of flagellar wrapping motility under physiological conditions. We have carefully addressed the points raised and provide detailed responses below.

*1. For the major experiment in Fig. 1, the motion of *C. insecticola* cells was observed in dissected symbiotic organs. Would this dissection affect the motion pattern of cells and alter the conclusions? For instance, was there flow in the CR, and if so, how would the flow affect the motion of bacterial cells? The authors could use tracers to examine the flow in the CR.*

We thank the reviewer for this important comment. We agree that flow in the dissected tissue could potentially influence bacterial motility. However, as shown in our previous study (Ref. 3), the constricted region (CR) is a highly narrow passage that is filled with mucin-like mucus and does not allow the passage of liquids such as colored water. This indicates that even under normal physiological conditions, the CR does not support bulk flow. In our current study, we carefully examined the dissected midgut tissues under microscopy and did not observe any visible flow that could influence bacterial motion. We have added the following sentences in L68-71 of the revised manuscript as “Since the CR is an extremely narrow passage through which even colored liquid cannot pass, as reported previously³, we consider that fluid flow is unlikely to occur in this region after dissection. Indeed, no visible flow was observed during our microscopic observations in the narrow passage.”

2. What accounts for the direction bias of cells towards the symbiont organ? In the Q-1D experiments, cells move persistently along either direction with wrapping motility, but within the insect body, cells must navigate towards the symbiont organ. Is there an asymmetry of the CR that could account for the direction bias?

We appreciate the reviewer's insightful question. We believe that the directional bias observed *in vivo* arises not from an intrinsic asymmetry of the CR structure but from the physical constraints imposed by its extreme narrowness. As seen in Movie S2, the cells enter the CR with their flagellar filaments positioned at the front. Once a bacterial cell enters the CR, it cannot reorient or turn around inside the passage. Therefore, the initial orientation at the point of entry determines the direction of motion, leading to a persistent movement toward the symbiotic organ. We have added the following sentences in L74-78 of the revised manuscript as "Notably, fluorescently labeled flagellar filaments were visualized at the front of the cell body during entry into the CR, indicating that cells orient themselves with their flagellar motors facing forward. Since U-turns are physically impossible once inside the narrow passage, this orientation plausibly leads to a directionally biased movement toward the symbiotic organ. This observation supports the idea that flagellar wrapping is important for breaking through the host's sorting organ."

3. In the Q-1D experiments, the authors use viscous solutions of 0.4% methylcellulose (MC). Is this solution a good model of CR mucus fluid? Measuring the viscoelasticity of CR fluid and comparing it with that of the MC solution would be helpful.

We appreciate the reviewer's thoughtful comment. To our knowledge, the viscoelastic properties of the CR fluid in *R. pedestris* have not been directly measured. This is primarily due to the extremely small size of second-instar nymphs, only a few millimeters in length, making it technically challenging to obtain sufficient amounts of CR fluid for rheological analysis. We are also conducting genetic analysis on mucin within CR, but we believe its identification will be the subject of subsequent studies. We have added the following sentence in L86-88 of the revised manuscript as "Although the exact viscoelastic properties of fluid in the CR are currently unknown, we used MC solution to approximate the physical parameters of the insect's sorting organ."

4. The authors' hydrodynamic modeling considers bacterial motion in a viscous fluid. However, CR mucus fluid and the MC solution used in the Q-1D experiments are both viscoelastic. Viscoelasticity of fluids is known to alter bacterial swimming behavior near solid surfaces. I wonder if any results would be modified if viscoelasticity is taken into account. Also, there is a recent numerical study on the propulsion of cells with wrapping motility (Gidituri et al, JFM 2024). Could the authors discuss the differences in their model?

We thank the reviewer for raising this important point. Indeed, both CR mucus and the MC solution used in our Q-1D device are viscoelastic fluids. However, we have performed an additional numerical calculation taking account for viscoelastic fluid and confirmed that the results are similar those reported in Newtonian fluids. We have added the following sentence in L196-198 of the revised manuscript as “Our hydrodynamic modeling considers bacterial motility in a viscous fluid; however, we confirmed that the results are qualitatively consistent with those in viscoelastic fluid (see Supplementary Information for details).” We have also added the detailed discussion in L153-176 of Supplementary Information.

We thank the reviewer for pointing out the recent study by Gidituri *et al.* We have cited the paper and added the following sentence in L196-199 of the revised manuscript as “Compared with a recent numerical study²¹, our model highlights a distinct regime in which the swimming speed of wrapped mode exceeds that of unwrapped mode in tightly confined spaces (see Supplementary Information for details).” We have also added the detailed discussion in L178-194 of Supplementary Information.

Minor points:

Line 18: Why would limited diffusion impair the movement of cells? Is the diffusion here referring to active diffusion?

We have modified the following sentence in L18 of the revised manuscript as “little is known about how bacteria behave under such conditions where movement is challenging due to spatial confinement that limited effective diffusion.”

Line 88: What is the relation between the cheA mutation and clockwise rotation of the motor?

We have added the following sentence in L100-102 of the revised manuscript as “Considering that the deletion mutant of *cheA*, which continuously rotates their flagellar motor in the counterclockwise (CCW), stopped movement in the device for a long time.”

Line 174: Hook bending stiffness was estimated by the thermal fluctuation of flagellar filaments. The authors should provide the details of the calculation or theory behind this important measurement.

We have added the following sentences in L575-579 of the revised manuscript as “The hook bending stiffness, EI , was estimated using the formula $EI = k_B T L_{hook} / \sigma^2$, where k_B is Boltzmann’s constant, T is the absolute temperature, a length of the hook L_{hook} (assumed to be 60 nm), and σ is the variances of the orientation angle of the flagellar filaments, as previously described^{12,25}. Assuming that the flagellar bundle in *C. insecticola* and *B. anthina* consist of three filaments, we defined the stiffness per hook as $A_{hook} \cdot EI/3$.

Reviewer #2 (Remarks to the Author):

In this manuscript, the authors report experimental results, supported by numerical modeling work, that demonstrate a key role of flagellar wrapping in navigating narrow channel structures as they are often encountered by motile bacteria. They mostly focus on the specific example of the stinkbug symbiont Caballeronia insecticola and present motility data both from the host's gut sorting organ and from PDMS-based microfluidic chips. They also extend their analysis to more than 10 related species and identify the mechanical properties of the hook as the key determinant of flagellar wrapping.

The manuscript is based on thorough experimental data, in particular the large numbers of cells that can be tracked in the linear microfluidic channels is a nice feature of this work, while the fluorescence imaging data seems a bit less abundant. Also the attempt to connect fundamental physical properties of the bacterial flagellum to biologically relevant questions related to insect symbionts are a nice feature of this work. However, several aspects remain unclear and require revision before publication in Nature Communications can be considered.

We sincerely thank the reviewer for the thoughtful and constructive comments on our manuscript. We greatly appreciate the recognition of the strengths of our work, including the comprehensive experimental dataset in more than 10 related species, the numerical modeling, and the effort to line the fundamental physical properties of the bacterial flagellum to biologically relevant questions in insect symbiosis. We have carefully considered the reviewer's suggestions and have revised the manuscript to address the points.

Major points:

- At the beginning of the section on "Flagellar wrapping gives an advantage in narrow passages" you show results on Salmonella in narrow channels. This is an interesting addition because of Salmonella's peritrichous flagellation compared to the polar flagellation of the other species discusses. However, the Salmonella part remains somewhat incomplete and unsatisfactory because no fluorescence data is included that shows how Salmonella actually operates their flagella to move in the channels. Do they switch from push to pull upon a change in direction? Or do they somehow reassemble their bundle on the other side of the cell body to reverse their direction of motion? This would be very interesting to know, as peritrichous bacteria normally do not move in pulling mode in open liquid. Why is no such data included? If labeling of the flagella failed for Salmonella, the authors could have chosen E. coli as the standard example of a peritrichous swimmer instead. I suggest to include such data.

We thank the reviewer for this insightful comment. In response, we have now included fluorescence imaging data of *Salmonella* with labeled flagellar filaments in the Q-1D device (Movie S11). We have added the following sentences in L159-166 of the revised manuscript as "To understand how cells reverse their direction of movement within the Q-1D, we first performed high-speed fluorescence

imaging of flagellar filaments in *Salmonella*, a peritrichous bacterium. During directional switching, the cell body retained its orientation, while the bundle of flagellar filaments at the rear unbundled and bundled again on the opposite pole, resulting in propulsion to the other direction (Movie S11). Notably, *Salmonella* cells consistently swam with their flagellar bundle trailing behind the cell body, regardless the direction of movement. We speculate that nanoscale gaps between the Q-1D and the glass surface allowed sufficient space for polymorphic transformation of the flagellar filaments.”

- Fig. 2c: I am confused about the trajectories shown in panel 2c. For a polarly flagellated swimmer, I would expect repeated reversals in swimming direction when the motor switches from CCW to CW and back (as for example in *V. alginolyticus*, *P. putida* and others). But here the trajectories look smooth, i.e. without reversals or other abrupt changes in swimming direction. Why are there no turns? In general: what does the swimming pattern of *C. insecticola* look like in open bulk liquid without any confinement? Is there no literature on this? I would expect that this is investigated before moving to confined environments.

We thank the reviewer for pointing this out. Indeed, *C. insecticola* cells do undergo directional reversals in bulk liquid, as expected for polarly flagellated bacteria. The trajectories shown in Fig. 2c were recorded in a relatively narrow field of view and over short time intervals, which may have given the impression of smooth, uninterrupted swimming. To clarify this point, we have now included trajectories of *C. insecticola* observed over a wider field of view and a longer time span for 4 s as a new Supplementary Figure (Fig. S2). These data clearly show repeated reversals during swimming in bulk liquid. While *C. insecticola* motility in open liquid has been reported in ref 13, our observations are consistent with the swimming pattern of polarly flagellated bacteria. We have added the following sentence in L88-90 of the revised manuscript as “In the bulk liquid, *C. insecticola* cells exhibited repeated directional reversals confirm that the shares the general reversal swimming pattern characteristic of polarly flagellated species (Fig. S2).”

- I am very confused about the directional bias reported in Fig. 2d. How do you define "forward" and "backward" directions? If your experimental setup (microfluidic device, distribution of cells, nutrients etc.) is symmetric, there is no reason why one of the directions along the microfluidic channel should be preferred, i.e., cells should, on average, move left or right with the same probability. Perhaps you have inserted cells only from one side, so that there is a gradient in medium concentration or quorum sensing factors or other chemical cues? From my understanding, the directional bias may be the signature of a chemotactic response (see, for example, Alirezaeizanjani et al, *Sci. Adv.* 2020; 6:eaaz6153, where the wrapped mode was identified as the chemotactically relevant swimming mode). Another possibility would be that all cell are entering the channels from one side with the motor ahead, so that there is a systematic difference between "forward" and "backward"? This aspect is not

addressed in the manuscript, not even in a speculative manner. You have to address and explain (or at least offer a plausible explanation) for the observation of the bias.

We appreciate the reviewer's thoughtful comment and agree that our original explanation lacked clarity. As pointed out, the microfluidic Q-1D device is fully symmetric, and there is no preferred directionality imposed by the experimental setup. The cells were not introduced into the channels from the open ends. Instead, we placed a suspension of the bacterial cell between the PDMS device and the glass coverslip, which passively confined the cells into the Q-1D channels during chamber assembly. We have added the following sentence in L549-552 of the revised manuscript as "Bacterial cells were not introduced from the open ends of the channels. Instead, a droplet of cell suspension was placed between the PDMS device and the glass coverslip, allowing cells to become passively confined within the Q-1D channels during assembly of the chamber."

In our analysis, "directional bias" was defined based on the net displacement of individual cells over a 1-minute period, regardless of whether they moved to the left or right. All movements, regardless of direction, were treated as positive values for the purpose of evaluating directional persistence, not absolute orientation. We have added the following sentence in L92-96 of the revised manuscript as "Since the Q-1D device is structurally symmetric and cells were randomly distributed throughout the channels, we measured the net displacement of individual cells, regardless of whether they moved to the left or right along the channel. In this analysis, the net displacement for 1 min in both directions were treated equally as positive displacements."

We also thank the reviewer for highlighting the potential link to chemotactic behavior. As noted in the study by Alirezaeizanjani et al. (Sci. Adv. 2020), wrapped-mode swimming may play a role in chemotactic responses. In our case, the directional persistence might reflect a behavioral strategy by the bacteria to escape from physically constrained environments where diffusion is severely limited, or possibly a chemotaxis-like response. We have added a discussion of this point in the revised manuscript. We have added the following sentence in L280-286 of the revised manuscript as "Interestingly, the observed directional persistence may reflect a physiological response of the bacteria to spatial confinement. As proposed in previous studies³⁴, wrapped-mode swimming has been associated with chemotactic behavior. In our case, the directional bias could indicate a chemotactic-like response, or alternatively, an intrinsic motility pattern that favors directional movement as a strategy to escape highly confined regions where chemical diffusion is extremely limited. Further studies will be needed to clarify the underlying mechanisms."

- Along the same lines: You show that cells move through the channels in wrapped mode. But what happens when they reverse their direction of motion? Do they stay in wrapped mode but simply move backward? Or do they unwrap their flagella and switch to push mode? Since you show movies with fluorescently labeled flagella in the channel, this information must be available. Please show

fluorescence movies, where cells can be seen that switch their direction of motion inside the channel.

We thank the reviewer for this important question. To address it, we analyzed high-speed fluorescence movies of *C. insecticola* with labeled flagellar filaments inside the Q-1D device (Movie S12). When the filaments were unwrapped, they trail behind the cell, and the cell propelled forward. Upon reversal of motor rotation, the cell moves in the opposite direction, with the flagellar filaments wrapped around the front of the cell body. We have added the following sentences in L166-172 of the revised manuscript as “In contrast, in *C. insecticola* (Movie S12), the flagellar filaments undergo a dynamic transition between wrapped and unwrapped configurations. When the filaments were unwrapped, they trail behind the cell, and the cell propelled forward. Upon reversal of motor rotation, the cell moves in the opposite direction, with the flagellar filaments wrapped around the front of the cell body. These findings demonstrate that in narrow passages, polar-flagellated *C. insecticola* exhibit direction-dependent flagellar configuration: they swim with wrapped flagellar filaments in one direction, while in the opposite direction, the flagellar filaments remained unwrapped.”

- *It would be helpful to be more explicit about the flagellation of the species you investigate. The statement on line 106 "To further investigate the importance of polar flagella..." suggests that they are all polarly flagellated. Is this true? And if yes, are they monotrichous or lophotrichous? Wrapping has been demonstrated for both cases but there may be still differences, see Park et al., Phys. Fluids 36, 101917 (2024).*

We thank the reviewer for this helpful comment. We agree that the flagellation patterns of the bacterial species examined should be clearly stated. In the revised manuscript, we now explicitly describe the types of flagellation as “*Caballeronia*, *Paraburkholderia*, and *Burkholderia* are either monotrichous or lophotrichous^{16,17}, *Pandoraea* are monotrichous¹⁸, whereas *Cupriavidus* are peritrichous¹⁹ (Table S1)”, in L128-129 of the revised manuscript. We hope that this clarification adequately addresses the reviewer’s concern.

- *The terminology (directional versus random) in the discussion of the MSD curves (Fig. S4b) is somewhat unclear/misleading. A parabolic MSD curve is a signature of ballistic motion ($x = v t$ and thus $x^2 = v^2 t^2$). But ballistic motion is not necessarily directional. If cells move ballistically in the channel, but with equal probability to the left and to the right, then the MSD will still be parabolic but the average displacement will remain zero, i.e. without directional bias. Similarly, from a linear in time MSD, you cannot conclude that there is no directional bias. To check whether a directional bias is present or not, I suggest that you also compute the mean displacement (not only the mean squared displacement), i.e. that displacement between $t=0$ and $t=t_{end}$ for each cell, averaged over all cells. This quantity will be non-zero if there is a directional bias.*

We thank the reviewer for this insightful comment. We calculated the mean displacement of cells,

defined as the average net displacement for each trajectory in Fig. S5c. We have added the following sentence in L116-119 of the revised manuscript as “To examine whether this ballistic motion was accompanied by a net directional bias, we further calculated the mean displacement (MD). *C. insecticola* and *V. fischeri* exhibited significantly positive MD values, whereas *S. enterica* cells displayed much smaller MD on average, suggesting relatively random movement with lower directional bias (Fig. S5c).”

- For those bacterial strains that do not move within the confined quasi-1D channel environment, such as the cheA deletion mutant or some of the related species that were additionally investigated, how were they introduced into the channels in the first place? In your movies, I can see that they are immobile but distributed all across the channels.

Thank you for raising this important point. We apologize for the lack of clarity. The cells were not introduced into the channels from the open ends. Instead, we placed a suspension of the bacterial cell between the PDMS device and the glass coverslip, which passively confined the cells into the Q-1D channels during chamber assembly. We have added the following sentence in L549-552 of the revised manuscript as “Bacterial cells were not introduced from the open ends of the channels. Instead, a droplet of cell suspension was placed between the PDMS device and the glass coverslip, allowing cells to become passively confined within the Q-1D channels during assembly of the chamber.”

- The modeling results showing that the hook stiffness is a key parameter for filament wrapping are not new and have been reported before, see Park et al. Scientific Reports 12, 6482 (2022), <https://doi.org/10.1038/s41598-022-09823-4>. Park et al. furthermore showed that not only the hook stiffness but, in addition, also the torque of the flagellar motor determines whether transitions to the wrapped mode occur, which is not even addressed here. The work of Park et al. has to be included and discussed here. Currently, I do not see any new insights from the hook modeling reported here that goes beyond what is already known from Park et al.

We thank the reviewer for this insightful comment. We now cite and discuss the study by Park et al. in the revised manuscript. As the reviewer correctly points out, their work highlights the importance of both hook stiffness and motor torque in determining transitions to the wrapped mode. While the present study focuses on the effect of hook stiffness under confinement, we agree that motor torque is also an essential factor. We are currently addressing this in a separate work that is under review. In addition, our model considers a broader range of flagellar filament morphologies, which provides complementary insights to those reported in Park et al. We have added the following sentences in L235-239 in the revised manuscript as “Previous modeling studies have demonstrated that both hook stiffness and motor torque play key roles in flagellar wrapping²⁷. While torque effects are not addressed in the present study, we are currently investigating this factor in a separate work²⁸. Our

simulations successfully reproduce diverse morphology of flagellar filaments observed in the experiments, thus providing complementary insights to those of the previous study²⁷.”

Smaller points:

- page 3: "...whereas the forward directional bias remained relatively consistent over time..." What do you mean by "consistent"?

We have updated the sentence as “whereas the forward directional bias remained nearly constant over time.”

- page 5: "...two distinct models" maybe you mean "modes" instead of "models"?

We have modified accordingly.

- page 5: "...no lateral suction flow..." How could there be such a lateral flow if there are the channel walls? How would it show up if there was such a flow?

We have modified the sentence as “we found that in the narrow tube, the unwrapped flagellum does not generate any significant flow from the lateral side of cell body.”

- page 5: What is ΔR ? Why do I need the cell length in this calculation? It is not clear what exactly is calculated here, please explain.

We have added the following sentences as “Here, ΔR represents the radial clearance between the bacterial surface and the channel wall. We defined the gap as normalized ΔR by $(H/2)$, where H is the cell length, to define a dimensionless parameter, facilitating comparison between different bacterial sizes.”

- Related to this: in the experiments you have rectangular channels but in your calculations, if I understand correctly, they are assumed to have a circular cross section. Does this make any difference? Please comment.

We have added the following sentences as “We modeled the channels as circular tubes for simplicity and numerical stability. Although the experimental microchannels have a rectangular cross-section, we expect the essential hydrodynamic behavior to remain qualitatively similar, as the dominant effect arises from the narrow confinement rather than the cross-sectional shape.”

- page 5: "estimated" In what sense is this an estimate? Isn't this rather the outcome of a numerical simulation?

We have updated the phrase as “calculated,” reflecting that this is the direct output of our numerical simulation.

- page 7: "inefficient rotation in the normal unwrapped mode" Where can I see this in the reported data?

Thank you for your comment. We have updated the following sentences in L271-273 of the revised manuscript as "this flexibility may potentially reduce rotation efficiency in the unwrapped mode due to reduced torque transmission. Although our experimental data do not directly measure this effect, it is plausible mechanical trade-off that warrants further study."

- Fig. 1g: Is there a way to improve quality of these panels? They are very difficult to read.

We have improved the figure quality by adding schematic overlays to each panel that indicate the cell body and flagellar filaments.

- Fig. 2f and Fig. S3d: How was this plot generated? Is there enough data to bin it in intervals of 0.5 sec and still have enough data in each time bin to reliably measure a mean run time? What is meant by "The ratio was sorted by 0.5s..."?

We apologize for the ambiguous phrasing. The plots were generated as follows: for each run, we measured its duration and classified it as either forward or backward. The ratio of forward runs was then calculated within duration intervals of 0.5 s (0–0.5 s, 0.5–1.0 s, etc.). The sample number of each bin are presented at the top. We have added the following sentence in the legend of Fig. 2f and Fig. S4d of the revised manuscript as "The duration of each run was measured, and the number of forward versus backward runs was counted. The ratio of forward runs was then calculated for runs with durations falling into 0.5 s intervals. The sample number of each bin are presented at the top."

- Fig. 3a: Where do I see the phylogenetic tree of *S. enterica* that you refer to in the caption?

We have removed the phrase of *S. enterica* from the caption in Fig. 3a.

- Fig. 3f: What are the grey parts in the horizontal bar plots?

We have added the following sentence as "Gray indicates uncategorized flagella that could not be clearly assigned to any of the defined categories."

- Fig. 3h: How was the gap width systematically changed? Can you control the width of the channel with such precision? I could not find anything about this in the methods section.

Thank you for your comment and for pointing out the need for clarification. We apologize for any confusion. The gap width presented in Fig. 3h was not experimentally varied but was systematically adjusted as a parameter in our numerical simulations. In these simulations, we defined the geometry of the narrow tube and systematically varied the gap between the cell surface and the channel wall to

examine its influence on swimming speed. We have added the following sentence in L492-494 of the revised manuscript as “The gap width between was defined geometrically and systematically varied in the numerical simulations to assess its effect on propulsion.”

- *Fig. 4f: I am not sure I understand the "competition index", maybe rephrase this?*

We have added the following sentences in L516-517 in the revised manuscript as “CI represents the relative colonization success of *C. insecticola* strains in the gut symbiotic organ. A higher CI indicates that *C. insecticola* with FlgE_{ci} outcompeted *C. insecticola* with FlgE_{Ba}.”

- *Fig. S3d: There seems to be a maximum forward bias at intermediate times. Any idea how to explain this or what this implies?*

Thank you for pointing this out. Overall, the forward bias remained consistent across different durations. As shown in Movie S11, peritrichous *Salmonella* cells maintain their cell body orientation while re-forming bundles on either side, which accounts for the lack of a systematic directional bias.

Reviewer #1 (Remarks to the Author):

I appreciate the authors' effort in addressing the comments. I still have the following concerns:

1. "However, as shown in our previous study (Ref. 3), the constricted region (CR) is a highly narrow passage that is filled with mucin-like mucus and does not allow the passage of liquids such as colored water. This indicates that even under normal physiological conditions, the CR does not support bulk flow. In our current study, we carefully examined the dissected midgut tissues under microscopy and did not observe any visible flow that could influence bacterial motion."

I am not fully convinced that there is no fluid flow in CR after checking Fig. 2 and Fig. 3 of Ref. 3. As the authors noted, the dye stained M1, M2 and M3 but never appeared in the M4B and the M4. If there is no flow in CR, the dye molecules will diffuse into the CR and develop a concentration gradient. However, Fig. 3A of Ref 3 showed a sharp concentration drop at the entry of the CR, which is highly unlikely if the molecules are diffusible and if there is no flow. One explanation for this fact is that there is flow (from M4 to M3 via the CR) flushing the molecules out of CR. Can the authors rule out this possibility? This question is critical because if there is fluid flow in CR, it would change the physical picture of *C. insecticola* navigation process, and the Q-1D microfluidic experiments as well as the hydrodynamic modeling would have to be modified.

We appreciate the reviewer's thoughtful comment and the opportunity to clarify this important point. We fully agree that the dye distribution pattern in Ref. 3 requires careful interpretation. However, we respectfully interpret that the apparent dye concentration gradient in the M3 region observed in Ref. 3 can be explained geometrically: the CR is conical, narrowing toward the posterior side, so that the local volume ratio changes sharply at its entrance, producing the appearance of a steep gradient even without convective flow. In addition, based on the digestive physiology of the bean bug *Riptortus pedestris*, we consider it highly unlikely that any fluid flow occurs either from M3 to CR or in the reverse direction from M4 to M3. As reported by Ohbayashi *et al.* (2015), this insect ingests liquid food by external digestion, which passes through M1, M2, and M3, where nutrients and water are fully absorbed, transported throughout the body via hemolymph, and metabolic wastes are collected by the Malpighian tubules (organs equivalent to kidneys) and excreted outside the body. When food coloring is ingested, hemolymph turn red dye to the food coloring absorbed in M1-M3 and are then excreted via the Malpighian tubules, but never appeared in M4B and M4, indicating that these posterior gut regions are functionally isolated from the digestive flow. This observation strongly supports that the CR and M4 compartments are not involved in bulk flow under normal physiological conditions. We have added the following sentences to the Results in L68-71 of the revised manuscript "*The bean bug R. pedestris* ingests liquid food that passes through M1, M2, and M3, where nutrients and water are fully absorbed, transported throughout the body via hemolymph, and metabolic wastes are excreted via the Malpighian tubules³. The posterior regions, including the CR, are therefore functionally isolated from the digestive flow."

2. *"To our knowledge, the viscoelastic properties of the CR fluid in R. pedestris have not been directly measured. This is primarily due to the extremely small size of second-instar nymphs, only a few millimeters in length, making it technically challenging to obtain sufficient amounts of CR fluid for rheological analysis. "*

Is it possible to perform microrheology measurement by feeding the bug with microspheres? The measurement can be done at M1, M2 or M3 since the fluid in these regions must be similar as that in the CR.

We agree that microrheology using fluorescent microspheres could, in principle, provide useful information about the viscoelastic properties of the gut fluid. However, the microspheres are unlikely to enter the CR. This inference is supported by the previous observation (Ref. 3 and 13) that non-motile symbiotic bacteria are also unable to enter the CR, indicating that only actively motile cells can traverse this narrow, mucus-filled passage. Measurements in the M1, M2, or M3 regions are technically feasible, and we are indeed planning to perform such experiments. Nevertheless, the present study focuses on bacterial motility in narrow spaces that mimic the CR environment, rather than on bulk rheology of the anterior gut fluid. Because the M3 and CR regions are structurally and functionally isolated, microrheology in M3 would not directly reflect the viscosity inside the CR. Still, we consider the M3 measurements valuable as a reference, and this work is currently being pursued as part of a separate study.

3. *I appreciate the authors' effort to perform additional numerical calculation taking account for viscoelasticity. But 0.4-0.5% methylcellulose solutions also display pronounced shear thinning viscosity. Instead of the Maxwell model, Giesekus fluid model would be more suitable. Anyhow, the calculation is based on the assumption that there is no fluid flow in the CR. If there is flow, the simulation setting would be substantially different.*

We thank the reviewer for raising the important point regarding shear-thinning behavior of methylcellulose and the suitability of constitutive models. In the present study, we adopted a Maxwell-type model to explore the qualitative effects of elasticity under the working hypothesis of negligible bulk flow in the CR. We have clarified this modeling choice to the Results in L188-190 as *"In this study, we employed linear Newtonian and Maxwell fluid models as minimal theoretical frameworks to discuss the swimming efficiency of the wrapping mode in confined environments."* We have also made our assumption and associated model settings explicit to the Results in L210-212 as *"We emphasize that our simulations were performed under the working hypothesis of a quiescent (no bulk-flow) CR environment. Under this assumption, our hydrodynamic modeling considers bacterial motility in a viscous fluid."* We plan to pursue these extensions in future work together with ongoing microrheological measurements of the midgut (M3).

My overall impression is that the current results in the manuscript have not yet demonstrated the biological and ecological functions of wrapping motility under physiological conditions, specifically within the CR tube of the bug. For this purpose one would have to first characterize the relevant physiological conditions, such as flow and rheology of CR fluids, and use Q-1D device to investigate the role of wrapping motility under those conditions. The current work with Q-1D microfluidic channels has its own right, but that seems more like a sequel of Ref. 13. The channel geometry of Q-1D experiment is relevant, but it is already well known that wrapping mode is beneficial for bacterial navigation in confined space.

We thank the reviewer for this important and constructive overall comment. We fully appreciate the desire to see a stronger demonstration of the biological and ecological relevance of wrapping motility under physiological conditions, particularly inside the host's constricted region (CR). Below we summarize how we have addressed these points in the revised manuscript, and we explain why we believe that the present work already provides new and biologically relevant evidence that flagellar wrapping facilitates translocation in one-micrometer-scale passages.

1. Novelty: direct demonstration that wrapping confers advantage in one-micrometer passages.

Although previous work has described the existence of flagellar wrapping, to our knowledge no prior study has directly demonstrated that wrapping provides a locomotory advantage specifically within microscale (~1 μm) constrained passages. In the previous version, we mentioned this point in Introduction as “*Since flagellar wrapping is known to be effective in moving in viscous conditions and escaping substrate surfaces¹⁵, this unique swimming mode is thought to play a pivotal role in adapting to the internal environments of the hosts². Little is known, however, about the ecological significance of flagellar wrapping in spatially-confined narrow passages.*” As the reviewer correctly pointed out, this wording did not make the novelty fully explicit. We have therefore added a clarifying statement to the Discussion in L276-279 of the revised manuscript as “*By combining in vivo imaging, Q-1D microfluidics, hydrodynamic modeling and genetic modification, our results are the first to demonstrate that confinement itself promotes wrapped configurations and that these configurations mechanically enhance translocation in one-micrometer-scale passages.*”

2. Relation to Ref.13 and the added value of the present study.

We have clarified the relationship with Ref.13 (Kinosita et al.), which reported flagellar wrapping in *C. insecticola* and *V. fischeri*, but did not establish that spatial confinement increases wrapping frequency or that wrapping enhances displacement in one-micrometer passages. We explicitly state this distinction to the Discussion in L273-276 of the revised manuscript as “*Importantly, whereas the phenomenon of flagellar wrapping in the symbionts of C. insecticola has been described previously¹³,*

to our knowledge no previous study has directly demonstrated that wrapping confers a locomotory advantage specifically within micrometer-scale constrictions.”

3. How Q-1D approximates physiological CR conditions.

We agree that physiological characterization is important. Rather than claiming exact identity, we show three independent lines of evidence that the Q-1D device reproduces key mechanical and transport constraints of the host CR. We have added a brief paragraph to the Results in L141-148 as *“Three lines of evidence support the relevance of the Q-1D device as a model for the host CR: (i) C. insecticola exhibits comparable net displacements in dissected CR–M4B tissue and in the Q-1D channels (Fig. 1e and Fig. 2d), (ii) the ability of closely related species to move efficiently in Q-1D positively correlates with their infection rates of the host (Fig. 3d), and (iii) the addition of methylcellulose in Q-1D reduces particle diffusion in a manner consistent with the mucus-like CR matrix (Fig. S1). Together, these results indicate that the Q-1D device captures essential mechanical and transport constraints relevant to the CR environment.”* We have also added the following sentence to Discussion in L279-282 of the revised manuscript as *“Although simplified, the Q-1D device reproduces key physical features of the CR, including geometric confinement, reduced effective diffusion, and comparable cell displacement, and thus provides an appropriate platform for examining how flagellar wrapping affects propulsion under host-relevant mechanical constraints.”*

We hope that these targeted clarifications address the reviewer’s concerns. The revisions explicitly highlight (i) the novelty of our direct demonstration that wrapping is advantageous in one-micrometer passages, (ii) how our study extends Ref.13, and (iii) why Q-1D provides a biologically relevant physical model of the CR. We sincerely appreciate the reviewer’s constructive comments, which have helped us to clarify the biological and ecological significance of our finding.

Reviewer #2 (Remarks to the Author):

I thank the authors for their thorough revision. All my questions and concerns have been addressed in great detail.

Overall, I am very happy with your revision, it is a beautiful paper and I can now recommend publication in Nature Communications.

Two points remain that I recommend to be addressed:

Following my request, you have added a new movie (S11) that shows a fluorescence microscopy recording of peritrichously flagellated Salmonella performing directional reversals in a channel. In the accompanying text, you say that you "speculate that nanoscale gaps between the Q-1D and the glass surface allowed sufficient space for polymorphic transformation of the flagellar filaments."

However, in the movie I can see that the flagellar bundle disassembles and spreads widely around the cell body before reassembling on the other side to drive motion in reverse direction. It seems that there is ample space of several micrometers around the cell body, not just "nanoscale gaps". It is a beautiful recording but is this really a channel of 1µm in width and depth? It seems much larger. I suggest, that you mark the channel walls in your movie (as they are not fluorescent, they are invisible in this recording). An if this is indeed a much wider channel, I suggest that you add more data that actually shows reversals of Salmonella in a 1µm squared channel.

We thank the reviewer for this careful observation. We have added the following sentence to the Result in L174-177 as *"In addition, a small gap may exist between the cell and the channel wall, allowing partial protrusion of the flagellar filaments. We regard this as a current technical limitation of device fabrication, and improving the precision of the microchannel geometry to minimize such nanoscale gaps will be an important direction for future development."* In the movies, the cell body remain confined within this narrow passage. To make this clearer, we have revised the supplementary Movies S11 and S12 by overlaying the channel walls in magenta. Each movie consists of two parts: the first half shows only the fluorescence signal, highlighting the flagellar dynamics during cell swimming along the channel; the second half presents the same sequence with the magenta overlay, indicating the channel boundaries. This format allows viewers to appreciate both the detailed flagellar motion and the spatial confinement of the cell within the 1 µm channel. We have also updated the figure captions.

Following another of my requests, you have added Fig. S5c showing the mean displacements. Am I correct in assuming that all these displacements were taken positive? So you actually show the average of the absolute value of the displacement $\langle |x| \rangle$, correct? This is simply the square root of the curves in S5b and does not add any new insight. What I actually had in mind was to show the average displacement taking its sign into account (displacements to the right with a positive sign and displacements to the left with a negative sign). If everything is isotropic, then these curves should be centered around zero (provided you have enough data). This would be a nice test to show that indeed everything is isotropic and there is no chemotactic or other gradient biasing. It would be a nice control and since you have all the data, it should be easy to compute.

I do not need to see the manuscript again and leave it to the editors to check that these points are fixed.

We thank the reviewer for this helpful suggestion. We fully agree that showing the signed mean displacement is a more appropriate way to verify isotropy in our Q-1D assays. In the revised version, we have replaced the previous Fig. S5c with new data presenting the signed mean displacement $\langle x \rangle$, where rightward motion is defined as positive and leftward as negative. The corresponding text and description of the former (absolute) mean displacement have been removed accordingly. The new analysis demonstrates that cell motion within the Q-1D device is isotropic. The mean displacement of

C. insecticola, *Salmonella*, and *V. fischeri* fluctuate symmetrically around zero, confirming the absence of any directional bias. Thus, no chemotactic or other gradient-driven effects are present in our microfluidic setup. We have also added the following statement to Results in L122-124 as "*In the Q-1D device, the signed mean displacement of C. insecticola, Salmonella, and V. fischeri moved symmetrically around zero, confirming that their motion was isotropic and not biased by chemotaxis or any other gradient (Fig. S5c).*"